# Microglia shape the embryonic development of mammalian respiratory networks

**Marie-Jeanne Cabirol[1†], Laura Cardoit[1†], Gilles Courtand[1], Marie-Eve Mayeur[2], John Simmers[1], Olivier Pascual[2], Muriel Thoby-Brisson[1]\***

[1]Institut de Neurosciences Cognitives et Intégratives d'Aquitaine, CNRS, Université de Bordeaux, Bordeaux, France; [2]MeLis INSERM U1314-CNRS UMR 5284, Faculté Rockefeller, Lyon, France

**Abstract** Microglia, brain-resident macrophages, play key roles during prenatal development in defining neural circuitry function, including ensuring proper synaptic wiring and maintaining homeostasis. Mammalian breathing rhythmogenesis arises from interacting brainstem neural networks that are assembled during embryonic development, but the specific role of microglia in this process remains unknown. Here, we investigated the anatomical and functional consequences of respiratory circuit formation in the absence of microglia. We first established the normal distribution of microglia within the wild-type (WT, $Spi1^{+/+}$ (Pu.1 WT)) mouse (*Mus musculus*) brainstem at embryonic ages when the respiratory networks are known to emerge (embryonic day (E) 14.5 for the parafacial respiratory group (epF) and E16.5 for the preBötzinger complex (preBötC)). In transgenic mice depleted of microglia ($Spi1^{-/-}$ (Pu.1 KO) mutant), we performed anatomical staining, calcium imaging, and electrophysiological recordings of neuronal activities in vitro to assess the status of these circuits at their respective times of functional emergence. Spontaneous respiratory-related activity recorded from reduced in vitro preparations showed an abnormally slow rhythm frequency expressed by the epF at E14.5, the preBötC at E16.5, and in the phrenic motor nerves from E16.5 onwards. These deficits were associated with a reduced number of active epF neurons, defects in commissural projections that couple the bilateral preBötC half-centers, and an accompanying decrease in their functional coordination. These abnormalities probably contribute to eventual neonatal death, since plethysmography revealed that E18.5 $Spi1^{-/-}$ embryos are unable to sustain breathing activity ex utero. Our results thus point to a crucial contribution of microglia in the proper establishment of the central respiratory command during embryonic development.

**\*For correspondence:**
muriel.thoby-brisson@u-bordeaux.fr

[†]These authors contributed equally to this work

## Editor's evaluation

This study presents fundamental experimental data from a mutant mouse model lacking microglia (Pu.1-/- mouse line) indicating that these cells have an important role in the embryonic establishment of critical neural circuits in the brainstem generating breathing motor behavior in mice. The authors examined in comparison to wild-type animals the anatomical and functional characteristics of two main respiratory neuronal groups-in the embryonic parafacial (epF) and the preBötzinger complex (preBötC) respiratory regions that operate together in the developing brainstem to generate the rhythmic neural activity necessary to establish normal breathing behavior and ensure survival at birth. Convincing experimental evidence is presented indicating that these respiratory networks become functional at typical developmental stages in the absence of microglia but exhibit anomalies in endogenous rhythm generation (slower respiratory rhythm). The authors' data suggest that these deficits are associated with reduced cell numbers of active neurons and abnormal rhythmogenesis

in epF and reduced commissural axonal projections affecting bilateral activity synchronization of the preBötC circuits generating inspiratory rhythm.

## Introduction

Microglia are brain-resident macrophages that invade the central nervous system (CNS) early during embryogenesis where they participate actively in brain homeostasis and in shaping neural circuit assembly. Indeed, it is now established that microglia play important roles in different aspects of the development of brain structures by controlling cell death, angiogenesis, synapse refining, neurogenesis, and axon tract formation (*Paolicelli et al., 2011*; *Thion and Garel, 2017*; *Li and Barres, 2018*; *Stevens and Schafer, 2018*).

Breathing behavior relies on several interacting rhythmogenic networks controlling motoneuronal pools located in specific regions of the brainstem and spinal cord. The two main brainstem neuronal groups involved in respiratory rhythmogenesis and motor burst activity phasing are the preBötzinger complex (preBötC) responsible for inspiration (*Smith et al., 1991*; *Gray et al., 2001*) and the para-facial respiratory group (pFRG), involved in pre-inspiration and active expiration, as well as in playing a crucial role in central chemoception (*Onimaru and Homma, 2003*; *Guyenet et al., 2019*; *Pisanski et al., 2020*). A third neuronal group, the post-inspiratory complex (Pico, *Anderson et al., 2016*) has been more recently identified, but was not considered in the present study since its embryonic origin and functional emergence have yet to be established. The initial formation of these circuits occurs early during development: in mice, an operational pFRG emerges first at embryonic day (E) 14.5 (hence named embryonic parafacial group (epF) at this stage; *Thoby-Brisson et al., 2009*) whereas the preBötC becomes active at E15.5 (*Thoby-Brisson et al., 2005*). From this latter stage onwards, these rhythmic bilateral networks are functionally and reciprocally connected, and the various respiratory motoneuronal pools are driven through complex ipsi- and contralateral projections (*Thoby-Brisson et al., 2005*; *Thoby-Brisson et al., 2009*; *Ruffault et al., 2015*; *Wu et al., 2017*).

The formation of functional networks requires their constitutive neurons to be born in specific anatomical niches before translocating to their final destination through active migratory processes, as well as the development of proper axonal extensions toward their targets and the establishment of appropriate synaptic connections. While the territories of origin for both epF and preBötC neurons are reasonably well documented (*Dubreuil et al., 2008*; *Dubreuil et al., 2009*; *Rose et al., 2009*; *Thoby-Brisson et al., 2009*; *Bouvier et al., 2010*; *Gray et al., 2010*), the accompanying developmental processes involved in establishing the functional respiratory command are not fully understood. In particular, the role that microglia might play in such early formative processes remains completely unknown.

Here, to investigate the extent to which microglia are implicated in the construction of operational respiratory networks during embryonic development, we examined the anatomical and functional status of the epF and preBötC networks during embryogenesis in a mouse line, the so-called $Spi1^{-/-}$ mutant (also called $Pu.1$ mutant; *Back et al., 2004*), which is devoid of microglia. To this end, we combined anatomical approaches with plethysmographic recordings of breathing behavior in vivo, electrophysiological recordings, and calcium imaging of fictive respiratory activity in reduced embryonic preparations (isolated brainstem or transverse brainstem slices) in vitro. By comparing data from wild-type (WT) and $Spi1^{-/-}$ mice embryos, our findings reveal for the first time that microglia make an important contribution to the prenatal development of the respiratory command necessary to ensure survival at birth.

## Results

### Microglia cell distribution in the developing embryonic hindbrain

In a first set of experiments, we assessed the presence and spatiotemporal organization of microglial cells in the brainstem of WT (Spi1$^{+/+}$) mice at different embryonic stages from E14.5 to E18.5, using immunostaining labeling against the specific microglial marker Iba1. Since the pia and choroid plexus (two other known Iba1-positive structures) were removed from our preparations, any positive staining detected was considered to be specific to microglia contained in central nervous structures. Microglia were found present in brainstem tissue throughout this developmental period (*Figure 1A, F, K*), with

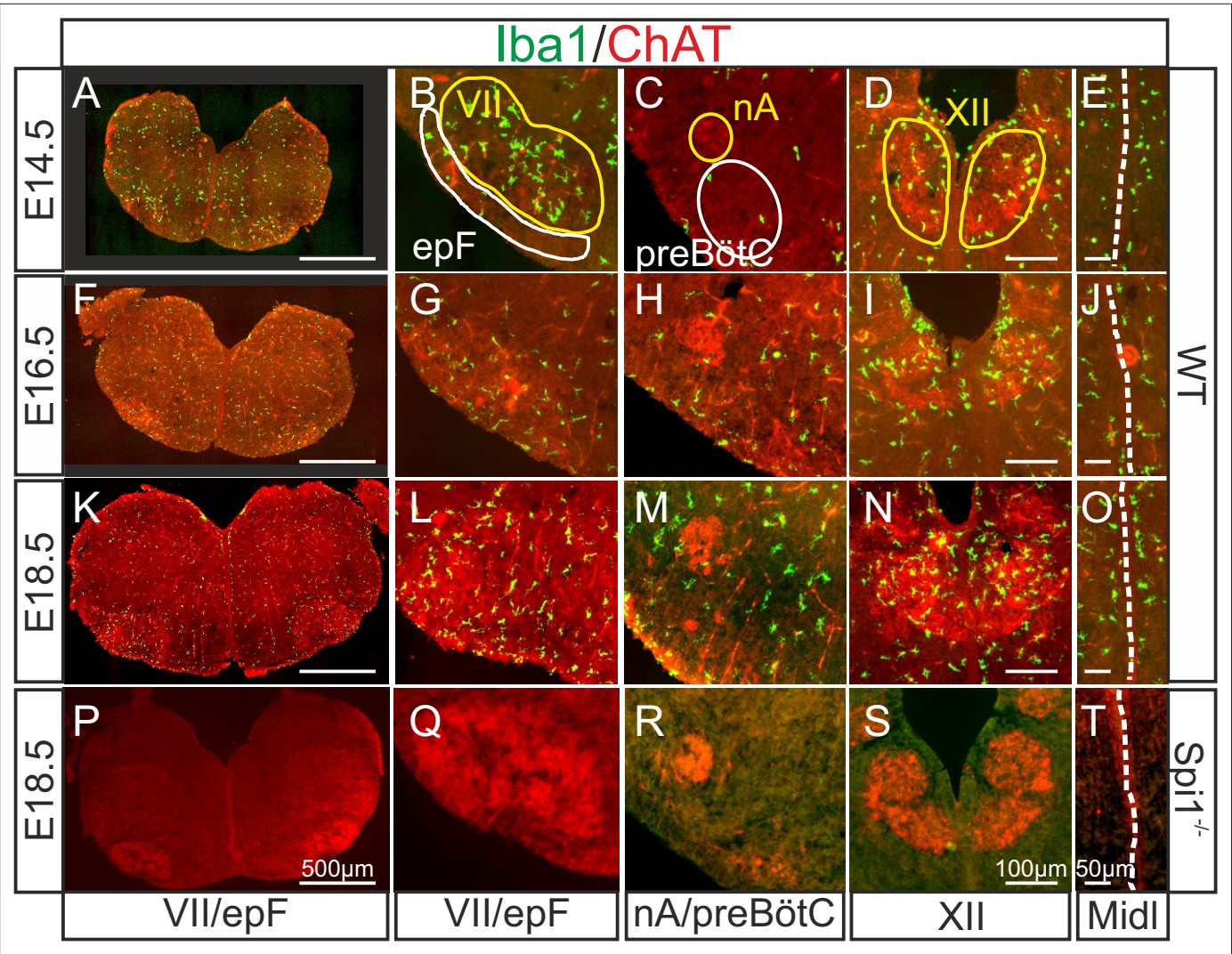

**Figure 1.** Immunodetection of microglia in the mouse hindbrain during embryonic development. Immunostaining for Iba1 (green) and ChAT (red) in transverse, 30 μm thick brainstem frozen sections obtained from three wild-type embryos (WT, upper three horizontal rows) and one Spi1$^{-/-}$ embryo (bottom row) at different embryonic ages (indicated on left) and at different axial levels (indicated at bottom). (**A, F, K, P**) Whole transverse brainstem slices shown at higher magnification at the level of the VII nucleus/epF (**B, G, L, Q**), the nucleus ambiguus/preBötzinger complex (**C, H, M, R**), the hypoglossal nucleus (**D, I, N, S**), and the midline (**E, J, O, T**). Yellow lines delineate brainstem motor nuclei, white lines delimit areas encompassing the respiratory networks of the epF and the preBötC. Note a preferential distribution of microglia in motor nuclei and along the midline, and the complete absence of Iba1 labeling in the *Spi1$^{-/-}$* mutant. epF: embryonic parafacial respiratory group; Midl: midline; nA: nucleus ambiguus; preBötC: preBötzinger complex; VII: facial motor nucleus; XII: hypoglossal nucleus.

an uneven distribution characterized by a preferential localization in motoneuronal pools, such as the facial (*Figure 1B, G, L*) and hypoglossal nuclei (*Figure 1D, I, N*). In contrast, selective concentrations of microglia in loci where the respiratory generators are located, ventral to the facial nucleus in the case of the epF (*Figure 1B, G, L*) and ventrolateral to the nucleus ambiguus for the preBötC (*Figure 1C, H, M*) were not observed. However, in all three developmental stages we detected a pronounced congregation of microglia flanking the two sides of the hindbrain midline (*Figure 1E, J, O*). This unexpected observation was of particular interest in the context of the bilaterally distributed rhythmogenic respiratory half-centers that are connected through commissural axonal pathways serving to synchronize the network activity of the two sides (*Thoby-Brisson et al., 2009*; *Bouvier et al., 2010*; *Wu et al., 2017*) (see below).

Next, we verified that microglia are indeed absent from the brainstem of the transgenic mouse line used in this study (*Back et al., 2004*) by performing immunostaining against Iba1 in brainstem

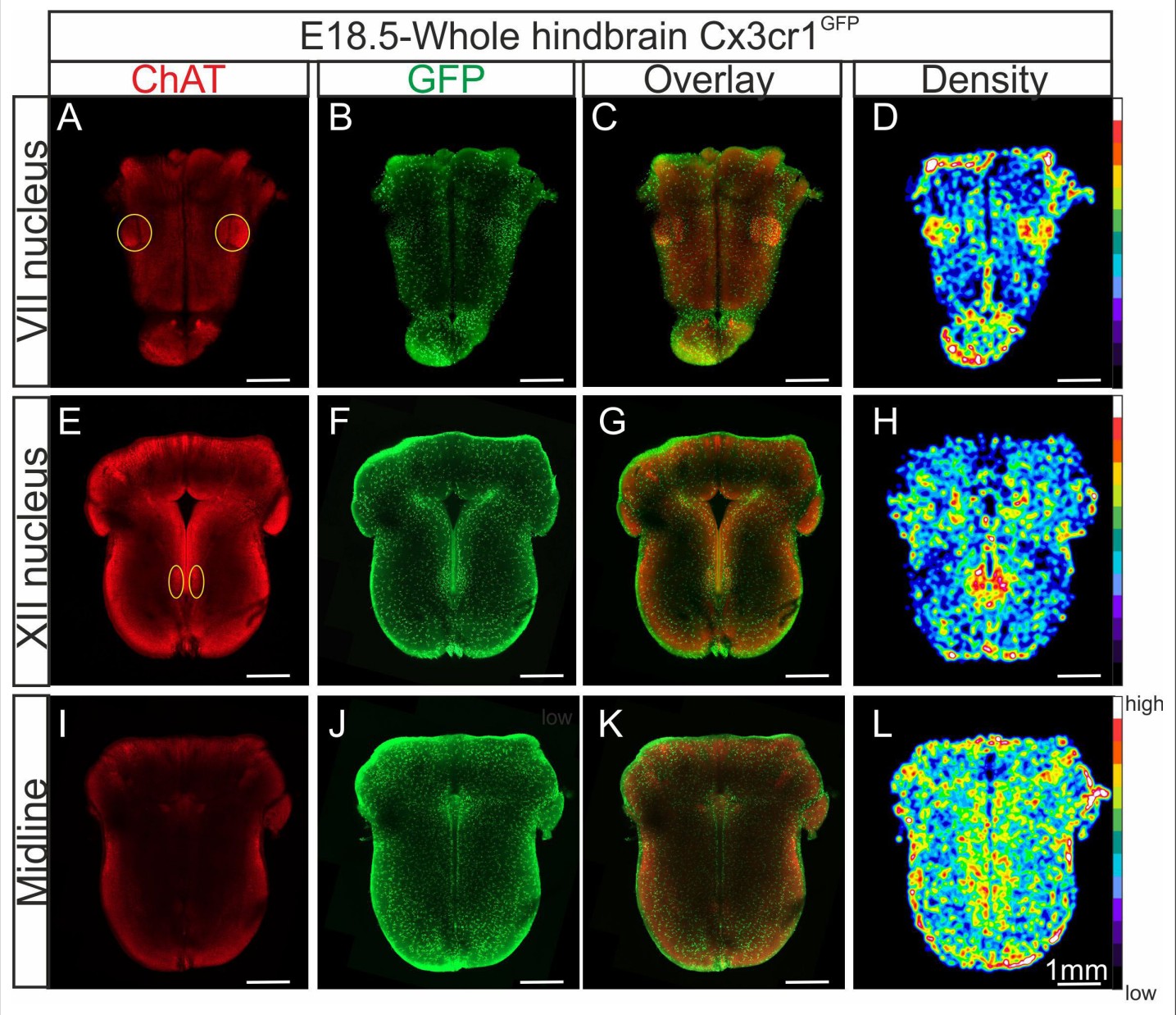

**Figure 2.** Distribution of microglia in a CLARITY-based cleared whole hindbrain obtained from a*CX₃CR-1^{GFP} embryo* at E18.5. Immunostaining for ChAT (red; **A, E, I**) and GFP (green; **B, F, J**) performed on hindbrain tissue with microglia endogenously expressing GFP. Images are orthogonal projections of optical sections stacks taken with a confocal microscope at the level of the facial nucleus (*n* = 18 images; top horizontal row), the hypoglossal nucleus (*n* = 12 images; middle row), and of the midline (*n* = 25 images; bottom row). (**C, G, K**) Overlay of ChAT and GFP labeling. (**D, H, L**) Density maps obtained from the GFP image stacks (see Materials and methods). Note the higher microglial cell densities in the region of the facial and hypoglossal nuclei and the midline. Yellow ovals highlight the positions of motor nuclei (facial nucleus in A and hypoglossal nucleus in E).

preparations obtained from *Spi1^{−/−}* embryos at E18.5 (*n* = 3). As expected, nervous tissue from such mutant embryos was completely devoid of labeling at all rostro-caudal levels examined (***Figure 1P–T***, see also Figures 4 and 5), including those regions in close proximity to brainstem motor nuclei and along the midline. These observations were thus consistent with a total lack of microglia in *Spi1^{−/−}* embryos.

In order to more accurately determine the distribution of microglia when present in the brainstem, we applied a tissue clearing protocol that allows 3D imaging of whole-hindbrain preparations obtained from *C*x3cr1^{GFP} E18.5 transgenic embryos in which microglia endogenously express GFP. With these preparations (*n* = 5), furthermore, we also performed immunolabeling for acetylcholine transferase (ChAT) to locate motoneurons (***Figure 2A, E***) conjointly with endogenous GFP labeling

(*Figure 2B, F, J*). Together with the subsequent construction of density maps from the GFP signal for microglia (*Figure 2D, H, L*), our data confirmed that higher concentrations of microglial cells are located at loci corresponding to the facial (*Figure 2D*) and hypoglossal nuclei (*Figure 2H*), as well as along the midline (*Figure 2L*).

## The epF network is anatomically and functionally altered in *Spi1⁻/⁻* embryos at E14.5

The onset of rhythmogenesis in the embryonic parafacial respiratory group (epF) is known to occur at E14.5 in the mouse (*Thoby-Brisson et al., 2009*). Neurons constituting this network are distinguished by their location (lateral and ventral to the facial nucleus), the expression of Phox2b, Atoh1, Egr2, and NK1R and by the absence of immunofluorescence with other anatomical markers (such as ChAT or Islet1,2) used to detect neighboring facial motoneurons (*Dubreuil et al., 2008*; *Pagliardini et al., 2008*; *Dubreuil et al., 2009*; *Rose et al., 2009*; *Thoby-Brisson et al., 2009*; *Ruffault et al., 2015*). Using whole-mount brainstem preparations (*Figure 3A*), as well as sagittal (*Figure 3B*) and coronal (not shown) slices obtained from WT ($n = 9$) and *Spi1⁻/⁻* ($n = 7$) embryos at E14.5, we performed immunostaining against Phox2b, NK1R, and Islet1,2 in order to locate cells comprising the epF network and to compare its anatomical integrity in the two genotypes. We observed that Phox2b⁺/NK1R⁺/Islet1,2⁻ expressing cells (i.e., epF neurons) located lateral and ventral to the Phox2b⁺/NK1R⁺/Islet1,2⁺ facial nucleus were significantly less numerous in the mutant compared to the WT (compare left and right *Figure 3A, B*). Specifically, from counts of all Phox2b⁺/Islet1,2⁻ cells in the epF region (ventral to the facial nucleus) of slices containing Phox2b⁺/Islet1,2⁺ facial motoneurons, the total number of epF neurons was found to be significantly decreased by 31% in the mutants, from a mean of 477 ± 36 cells in WT preparations ($n = 6$) to 285 ± 19 cells in *Spi1⁻/⁻* preparations ($p = 0.002$; $n = 5$; *Figure 3C*). In addition, we also observed a significant reduction in the rostro-caudal extension of the facial nuclei in *Spi1⁻/⁻* preparations compared to WT preparations, from 436 ± 7 µm in WT ($n = 11$) to 380 ± 8 µm in *Spi1⁻/⁻* hindbrains ($n = 11$; $p < 0.001$; *Figure 3D*).

We then asked whether such a substantial anatomical alteration has any functional consequences for the operation of the mutant's epF network. To address this question, we performed multiple cell calcium imaging on the ventral surface of isolated hindbrain preparations at E14.5 to detect spontaneously active epF cells, as described previously (*Dubreuil et al., 2009*; *Thoby-Brisson et al., 2009*; *Ramanantsoa et al., 2011*). In eight WT preparations, a column of cells generating rhythmically organized fluorescence fluctuations corresponding to neuronal impulse burst activity and occurring at a mean frequency of 12.4 ± 0.9 calcium transients/min were recorded in the region of the epF (*Figure 3E, F*). Moreover, when preparations ($n = 4$) were superfused by an artificial cerebrospinal fluid (aCSF) solution with a lowered pH (decreased from 7.4 to 7.2), the cycle frequency of ongoing fluorescence signals increased to 18.7 ± 1 calcium transients/min ($p = 0.002$), as expected due to the known intrinsic sensitivity of the epF network to acidosis (*Figure 3F*; *Thoby-Brisson et al., 2009*). In contrast, in mutant preparations under normal aCSF ($n = 8$), and consistent with our anatomical data (*Figure 3A, B*, right), functional imaging revealed that a significantly lower number of rhythmically active epF cells could be detected (*Figure 3E*, right). In addition, the spontaneous epF network rhythm monitored under control aCSF was generated at a significantly reduced cycle frequency compared to WT control of 8.2 ± 0.7 calcium transients/min ($p = 0.003$; *Figure 3F*). On the other hand, the network's sensitivity to pH changes was conserved in the mutant, since reducing the aCSF pH to 7.2 induced a significant frequency increase to 14.4 ± 1.5 calcium transients/min ($p = 0.001$; *Figure 3F*). Together these findings indicate that the absence of microglia during embryonic development leads to both anatomical and functional abnormalities in the epF network at E14.5, although its chemosensitivity to acidosis is maintained.

## PreBötC rhythmogenesis is preserved in *Spi1⁻/⁻* embryos at E16.5 despite abnormal commissural projections

The second major respiratory network, the preBötC starts to become active at E15.5 in the mouse and is fully functional by E16.5 (*Thoby-Brisson et al., 2005*). While the anatomical status of the preBötC remains impossible to investigate due to the current lack of network-specific markers associated with its heterogeneous composition (*Feldman et al., 2013*), its overall functional status can be tested by monitoring spontaneous activity with a macroelectrode placed on transverse brainstem slices in which

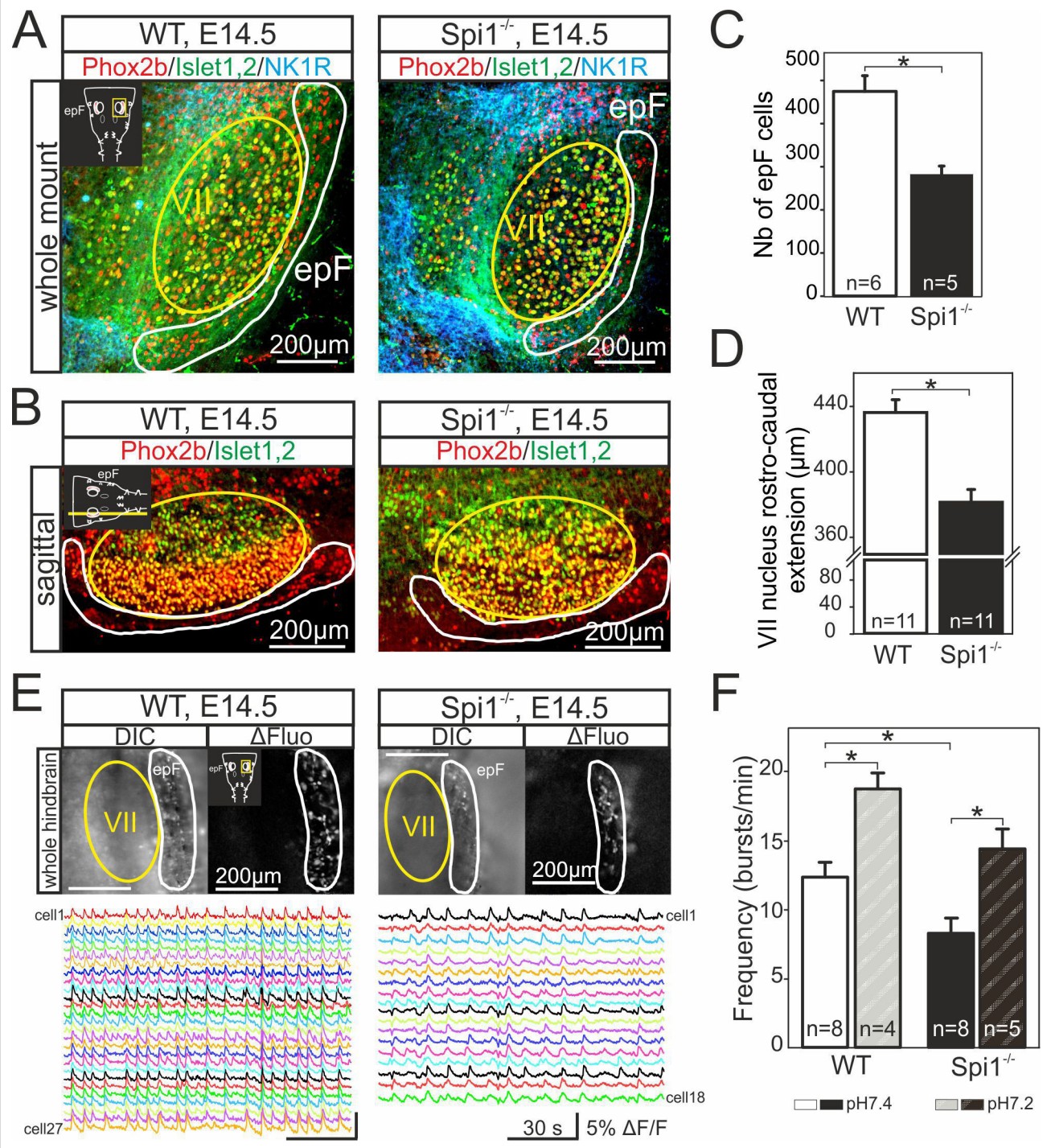

**Figure 3.** Anatomical and functional anomalies of the epF in *Spi1⁻/⁻* embryos at E14.5. (**A**) Partial ventral view of a whole-hindbrain preparations obtained from wild-type (WT, left panel) and *Spi1⁻/⁻* (right panel) embryos at E14.5 after triple immunolabeling with antibodies specific to Phox2b (red), Islet1,2 (green), and NK1R (blue). Cells of the epF express Phox2b and are located in a NK1R-positive region, ventral and lateral to facial cells that express Phox2b and Islet1,2. (**B**) Single sagittal slices from WT and *Spi1⁻/⁻* embryos at E14.5 after immunolabeling against Phox2b and Islet1,2. Insets in left A and B are schematic representations of hindbrain preparations in the orientation illustrated in the photographs. (**C**) Quantification of the total number of epF cells (Phox2b⁺/islet1,2⁻) in WT (unfilled bar) and *Spi1⁻/⁻* (black bar) preparations. The epF network in the mutant contains a significantly lower number (p < 0.01) of constituent neurons compared to WT. (**D**) Rostro-caudal extension of the VII nucleus in WT (unfilled bar) and *Spi1⁻/⁻* (black bar) preparations. (**E**) Images of Calcium Green 1AM-loaded whole-hindbrain preparations taken in DIC (left panels) and fluorescence (right panels) modes showing epF cells located lateral to the facial nucleus. Sample recording traces below illustrate repetitive fluorescent changes occurring simultaneously in individual epF cells. (**F**) Bar graphs quantifying the frequency of the fluorescent transients in WT and *Spi1⁻/⁻* preparations in control

*Figure 3 continued on next page*

*Figure 3 continued*

pH 7.4 (unfilled and black bars, respectively) and after acidification (pH 7.2) (dashed bars). Values are given as mean ± SEM. Numbers in bars indicate numbers of preparations analyzed. Student's *t* test and Mann-Whitney Rank Sum test have been performed. Asterisks indicate p < 0.01. epF: embryonic parafacial respiratory group; VII: facial nucleus. The epF network in the *Spi1⁻/⁻* mutant is composed of fewer cells that generate a slower rhythm compared to WT, although the network's chemosensitivity is maintained.

the neuronal domains of the preBötC are anatomically conserved. Such population recordings from E16.5 WT (*n* = 6) and *Spi1⁻/⁻* (*n* = 7) slices (*Figure 4A, B*) revealed the spontaneous occurrence of rhythmic burst activity at the locus of the preBötC network in all preparations tested, irrespective of genotype. Here again, this activity was generated at a significantly reduced cycle frequency in preparations from *Spi1⁻/⁻* embryos (5.5 ± 0.4 bursts/min) compared to 7.6 ± 0.3 bursts/min (p < 0.001) in WT preparations (*Figure 4B, C*), whereas the regularity of burst cycle periods remained comparable as indicated by the similar coefficients of variation in the two genotypes (*Figure 4D*). However, in

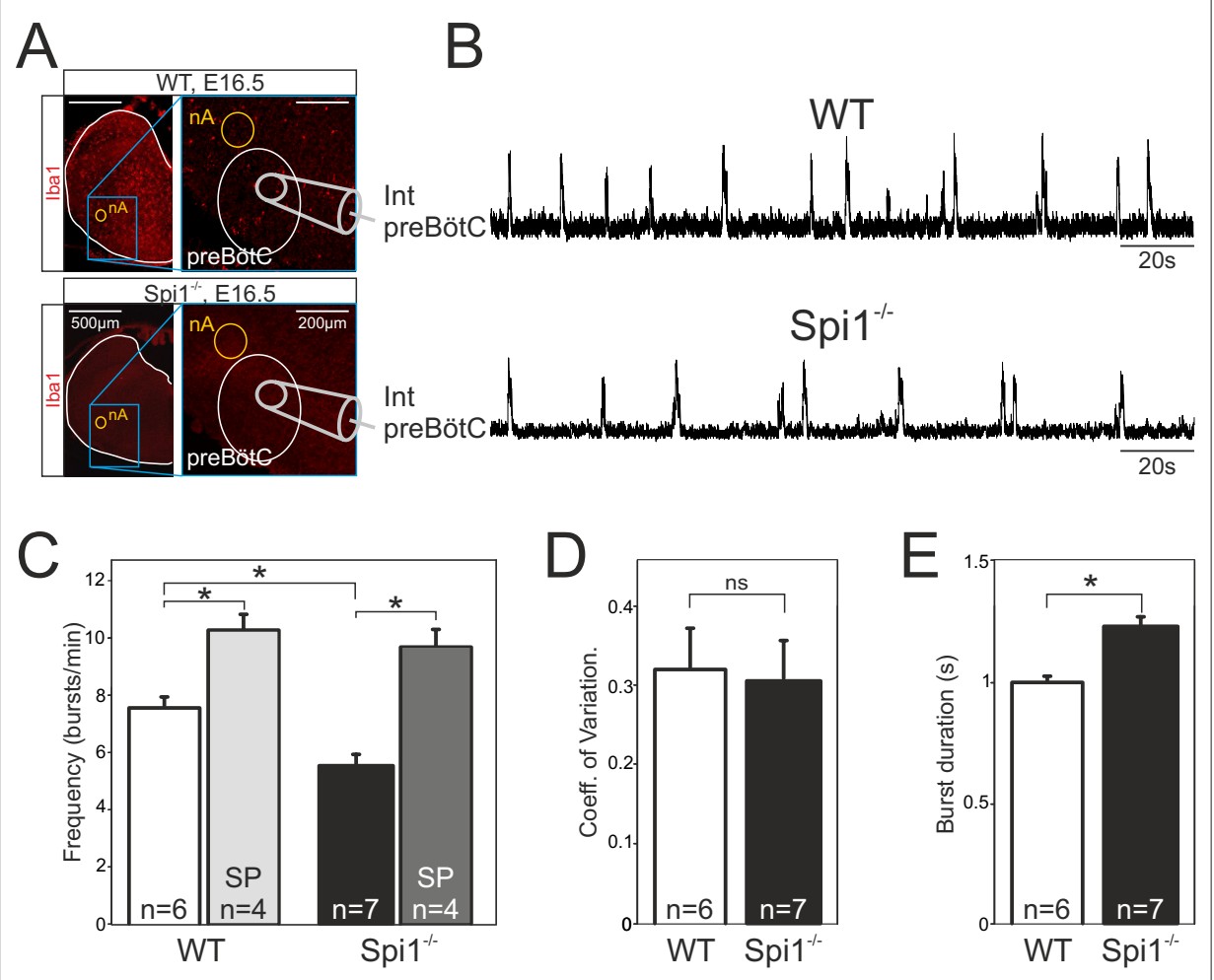

**Figure 4.** The preBötzinger complex (preBötC) network generates a slow rhythm with longer inspiratory bursts in the *Spi1⁻/⁻* embryo at E16.5. (**A**) Photomicrographs at two different magnifications of E16.5 transverse medullary slices from wild-type (WT) (top) and *Spi1⁻/⁻* (bottom) isolating the preBötC network and immunostained with the antibody against Iba to detect microglial cells. Note the complete absence of labeling in the *Spi1⁻/⁻* preparation. Also schematically represented is the position of the electrode used for population recording from the preBötC. (**B**) Integrated neurogram recordings from the preBötC network in a WT (top) and a *Spi1⁻/⁻* (bottom) slice preparation. (**C**) Quantification of the mean frequency of burst activity in the preBötC networks of WT (unfilled bars) and *Spi1⁻/⁻* (black bars) in control conditions and in the presence of 0.5 µM Substance P (gray bars). Same group recordings as analyzed in C showing mean coefficients of variation of the timing of bursts (**D**) and mean burst durations (**E**) of preBötC activity in WT and *Spi1⁻/⁻* embryos. Values are given as mean ± SEM. Student's *t* test have been performed. Asterisks indicate a p < 0.001. *n*, number of slices analyzed; nA: nucleus ambiguus; SP: Substance P. In the *Spi1⁻/⁻* the preBötC network generates respiratory activity at a lower frequency with longer bursts.

addition to a lower frequency we also found a significant increase (p < 0.001) in the mean values of burst durations that varied from 1.23 ± 0.03 s in preparations obtained from $Spi1^{-/-}$ embryos (n = 7) compared to 1.00 ± 0.02 s in WT preparations (n = 6) (*Figure 4E*).

In a number of preparations, we also assessed the sensitivity of the preBötC network to Substance P (SP), a well-known and potent modulatory peptide of this respiratory oscillator's operation (*Gray et al., 1999*; *Baertsch and Ramirez, 2019*). Exposure to 0.5 µM SP (as commonly used on such reduced embryonic preparations; *Thoby-Brisson et al., 2005*; *Chevalier et al., 2016*) induced a significant increase (p < 0.001) in the preBötC's rhythm frequency in both genotypes to 10.2 ± 0.6 bursts/min in WT preparations (n = 4), and to 9.6 ± 0.6 bursts/min in $Spi1^{-/-}$ preparations (n = 4) (*Figure 4C*). Overall these findings therefore indicated that despite generating a slower (albeit regular) burst rhythm composed of longer bursts, the mutant's preBötC network is clearly functional at E16.5 and remains sensitive to a major peptidergic neuromodulator after having developed in a tissue environment devoid of microglia.

Given the predominance of microglia along the brainstem midline (see *Figures 1 and 2*), microglia's known regulatory role in axon tract formation in the embryonic brain (*Pont-Lezica et al., 2014*; *Squarzoni et al., 2014*), and the bilateral distribution of the two preBötC neuronal groups in rhythmogenic half-centers interconnected and synchronized via commissural axonal pathways (*Bouvier et al., 2010*), we asked whether the developmental establishment of these midline-crossing projections might be subject to microglial control. More precisely, was there evidence for a disruption of the commissural pathways between the two preBötC half-centers in $Spi1^{-/-}$ mutants, and if so, how might this interfere with overall preBötC network function? To address this issue, we first labeled commissural projections from the preBötC network on one side by the unilateral application of a fluorescent dye crystal (Alexa Fluor 488 dextran) to WT (n = 7) and $Spi1^{-/-}$ (n = 6) slice preparations taken from E16.5 to E18.5 embryos. The midline was unequivocally localized by its strong immunoreactivity to the anti-NK1R antibody, as previously reported (*Thoby-Brisson et al., 2005*). As illustrated in *Figure 5A* (four examples of each genotype), considerably less commissural fibers were found to traverse the hindbrain midline in mutants compared to WT (*Figure 5A1, A2*). We then assessed whether such an anatomical impairment of commissural connectivity in the mutant had functional consequences for the coordination between the preBötC networks of each side. Simultaneous macroelectrode recordings were therefore made from both the left and right preBötC networks in WT (n = 7) and $Spi1^{-/-}$ (n = 8) preparations between E16.5 and E18.5, as for a typical WT preparation illustrated in *Figure 5B1*. From measurements of the delay between the onset of a given preBötC network burst and that of its contralateral partner, in contrast to the strict synchrony of burst pairs in WT preparations, a significantly wider range of delays was observed in $Spi1^{-/-}$ preparations, with mean values attaining 48.9 ± 27.8 ms in the latter preparations compared to −4±11.7 ms in the former (*Figure 5B2, B3*; 15 bursts per preparation; p < 0.05). This greater variability in the coordination between corresponding left and right preBötC bursts of activity probably also partially explains their longer durations (see Discussion). Thus, in the absence of microglia during embryonic development, commissural projections between the bilateral preBötC networks develop abnormally, likely underlying the observed weakened synchronization between the two oscillator half-centers, and in turn, to the generation of longer and less frequent bursts of respiratory-related activity.

## Phrenic motor activity is affected in $Spi1^{-/-}$ mouse embryos at E18.5

From E15.5 onwards, the epF and preBötC networks interact to produce the central respiratory command sent to the motoneuronal pools that drive breathing movements (*Thoby-Brisson et al., 2009*). The main intervening targets are phrenic motoneurons located in the rostral region of the spinal cord and whose axons exit in the phrenic nerves to innervate the diaphragm muscles. In order to assess whether phrenic motor output arising from the combined operation of the epF and preBötC networks is itself affected in $Spi1^{-/-}$ embryos, we recorded from phrenic motor axons carried in a fourth cervical root (C4) of isolated brainstem preparations at E18.5. We chose this embryonic age in order to examine the respiratory command output at a stage directly preceding birth when it has become fully operational in order to sustain survival at birth. Rhythmic motor bursts were detected in all 11 WT and 7 $Spi1^{-/-}$ preparations tested (*Figure 6A*). However, such fictive respiratory rhythms in mutant preparations were still generated with a significantly lower frequency than those recorded from WT preparations ($Spi1^{-/-}$, 10.6 ± 0.5 bursts/min; WT, 14 ± 0.4 bursts/min, p < 0.001), (*Figure 6B1*), although

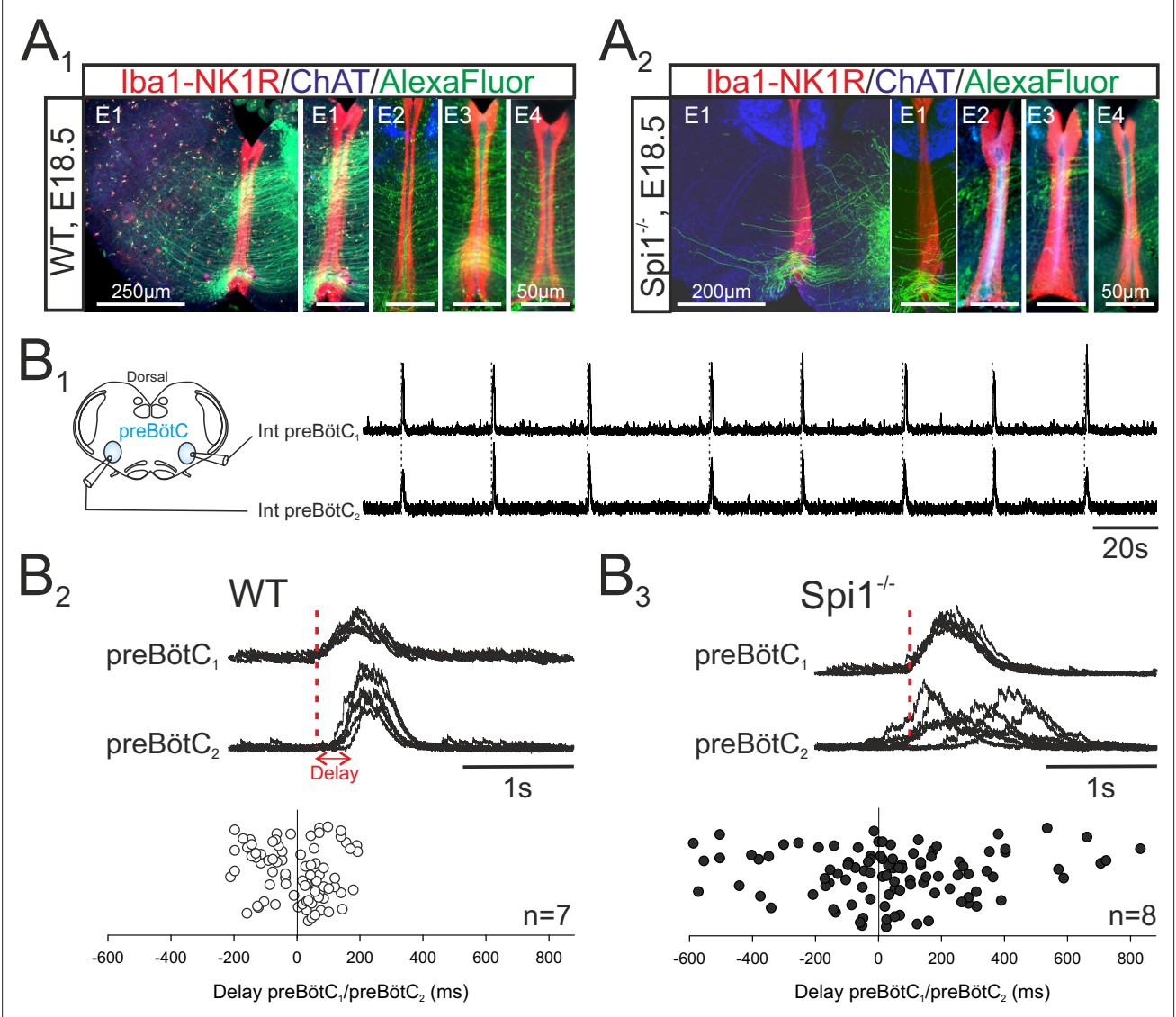

**Figure 5.** The bilateral preBötzinger complex (preBötC) networks exhibit abnormal commissural projections and diminished inter-network synchronization in the $Spi1^{-/-}$ embryo at E16.5–E18.5. (**A**) Photomicrographs of transverse brainstem slice preparations immunolabeled for Iba1 and NK1R (red), ChAT (blue), and with commissural projections from one preBötC network to its contralateral partner labeled with Alexa Fluor 488 (green) for wild-type (WT) (**A1**) and $Spi1^{-/-}$ (**A2**) preparations. Commissures crossing the midline are illustrated at a higher magnification for four different preparations (E1–E4) for each genotype. (**B1**) Simultaneous integrated recordings of bursting activity in bilateral preBötC networks of a transverse brainstem slice from a WT preparation (schematically represented at left) at E16.5. Dashed lines indicate time-related bilateral bursts. (**B2**) Superimposed integrated traces ($n = 7$) of recordings of burst activity occurring simultaneously in the two preBötC networks of a WT preparation. The red dashed line indicates the onset of bursts in preBötC$_1$ to which the superimposed traces were aligned. Bottom: Distribution of the delays between the onsets of bursts in the preBötC$_1$ and preBötC$_2$ networks. Values were obtained from 15 consecutive bursts in each of seven different preparations. (**B3**) Same arrangement as in B2 for $Spi1^{-/-}$ preparations ($n = 8$). Bursts on both sides are strictly time-locked in the WT but much less synchronized in $Spi1^{-/-}$ embryos at E16.5–E18.5.

again with a comparable rhythm regularity (p = 0.7; *Figure 6B2*). In addition, individual phrenic motor bursts exhibited longer durations in the mutant preparations (948 ± 18 ms) compared to WT (843 ± 11 ms, p < 0.001), (*Figure 6B3*).

We also tested the responsiveness of brainstem preparations to aCSF acidification that, due to the specific intrinsic chemosensitivity of the epF network and the latter's entraining influence on the preBötC network, is normally reflected by pH-dependent changes in the central respiratory drive to phrenic motoneurons (*Thoby-Brisson et al., 2009*). Accordingly, in control WT preparations (n = 6), ongoing rhythmic motor bursts recorded from C4 roots were accelerated by lowering the pH (from

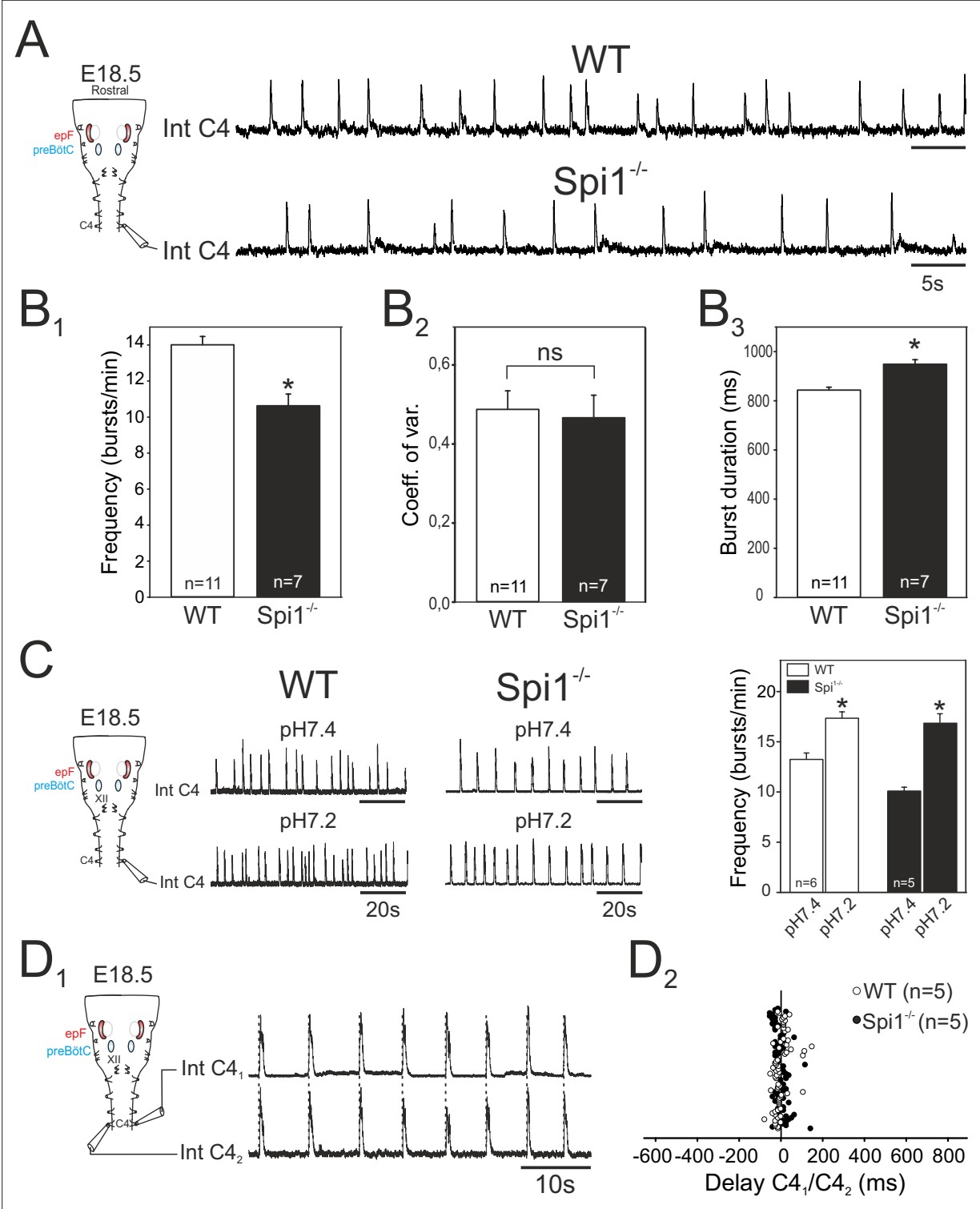

**Figure 6.** Characterization of respiratory-related activity generated by the isolated hindbrain of *Spi1⁻/⁻* embryos at E18.5. (**A**) Left: Schematic representation of the isolated brainstem preparation showing the position of the electrode used to monitor respiratory-related motor activity in the C4 phrenic motor root. Right: Integrated phrenic nerve discharge (Int C4) obtained in a preparation from wild-type (WT) (top) and *Spi1⁻/⁻* (bottom) embryos at E18.5. (**B**) Quantification of mean phrenic burst frequency (**B1**), coefficient of variation (**B2**), and burst duration (**B3**) for 11 WT (unfilled bars) and 7 *Spi1⁻/⁻* (black bars) preparations. (**C**) Same layout as in A for activities recorded under artificial cerebrospinal fluid (aCSF) at pH 7.4 (top traces) and at pH 7.2 (bottom traces) for WT (left traces) and *Spi1⁻/⁻* (right traces) preparations. Right: Quantification of mean phrenic burst frequency in pH 7.4 and pH 7.2 for six WT (unfilled bars) and five *Spi1⁻/⁻* (black bars) preparations. (**D1**) Integrated phrenic neurograms (IntC4) recorded simultaneously from the

*Figure 6 continued on next page*

*Figure 6 continued*

right and left C4 roots of an isolated hindbrain preparation at E18.5. Vertical dashed lines highlight the left/right synchrony of timely related bursts. (**D2**) Distribution of the delays between the onsets of bursts recorded simultaneously in the bilateral phrenic roots obtained from 15 consecutive bursts in five WT (unfilled dots) and five *Spi1⁻/⁻* (black dots) preparations. Left and right phrenic bursts are strictly time-coupled in both genotypes. Values are given as mean ± SEM. Student's *t* test have been performed. Asterisks indicate p < 0.001; ns: p > 0.1; *n*, number of preparations analyzed. epF: embryonic parafacial respiratory group. The motor output of the central respiratory command recorded from C4 roots was generated at a lower frequency and with longer bursts in the *Spi1⁻/⁻* mutant compared to WT, but remained chemosensitive and bilaterally coordinated.

7.4 to 7.2) of the bathing aCSF from 13.3 ± 0.6 to 17.4 ± 0.6 bursts/min (p < 0.001; *Figure 6C*, left). Similarly, however, in mutant littermates (*n* = 5), an equivalent aCSF acidification also induced a significant increase in phrenic burst frequency from 10.1 ± 0.3 to 16.9±0.9 bursts/min (p < 0.001, *Figure 6C*, right). This finding therefore demonstrated that despite a reduced frequency and a diminished epF network, the final respiratory drive to phrenic motoneurons continues to express centrally detected pH changes in the *Spi1⁻/⁻* embryos at E18.5, further confirming that the central chemosensitive property of the epf network is preserved.

In an ultimate series of in vitro experiments we asked whether, when operating together when fully developed at E18.5, the coupled epF and preBötC networks remain capable of producing an effective, bilaterally synchronous respiratory command in the *Spi1⁻/⁻* mutant, despite poorly developed preBötC commissural connectivity, and possibly midline-crossing defects at other brainstem and spinal cord levels (*Wu et al., 2017*). Simultaneous recordings from both left and right C4 roots in individual mutant brainstem preparations (*Figure 6D1*) unexpectedly revealed a persistence of synchronized respiratory-related bursting in spinal phrenic motoneurons on the two sides (*Figure 6D2*). Indeed, bursts of activity in one C4 root were strictly coordinated with those of the contralateral root, with a mean delay of 0.8 ± 4 ms in the *Spi1⁻/⁻* mutant (range from −51.8 to 142.4 ms; *n* = 5, 14 bursts per preparation), similar (p = 0.6) to that observed in WT preparations (mean: 2.07 ± 4.6 ms; range from −77.6 to 149.1 ms; *n* = 5, 15 bursts per preparation). Thus, the bilateral synchronization of the central respiratory command to the two phrenic motor pools that derives from the epF and preBötC networks acting in combination is apparently unaffected by the absence of microglia in the *Spi1⁻/⁻* mutant at the late prenatal stage.

## Lethal failure of breathing in Spi1 mutants at E18.5

On the basis of the respiratory network abnormalities revealed in our in vitro experiments, in a final set of experiments we assessed the consequences of an absence of microglia for actual breathing in vivo. Since *Spi1⁻/⁻* embryos die very rapidly after birth (*Back et al., 2004*, personal observation) we measured the breathing activity of E18.5 embryos that were delivered by caesarian section and manually stimulated until the induction of breathing behavior. Plethysmographic recordings of breathing activity was performed on 15 WT and 11 *Spi1⁻/⁻* embryos. Of the WT embryos, 14/15 animals expressed sustained rhythmic breathing movements in response to mechanical stimulation at a mean frequency of 102.4 ± 3 breaths/min (*Figure 7A*), whereas the remaining WT embryo failed to do so for an unknown reason. In direct contrast, the vast majority (9/11) of the *Spi1⁻/⁻* embryos, although they initially expressed a heartbeat, never began to generate breathing movements as evidenced by a virtually flat trace in the plethysmograph recordings (*Figure 7B*). The remaining two mutant embryos expressed abnormal breathing behavior for only a few minutes but died rapidly hereafter. Therefore, the anomalies that emerge in the central respiratory

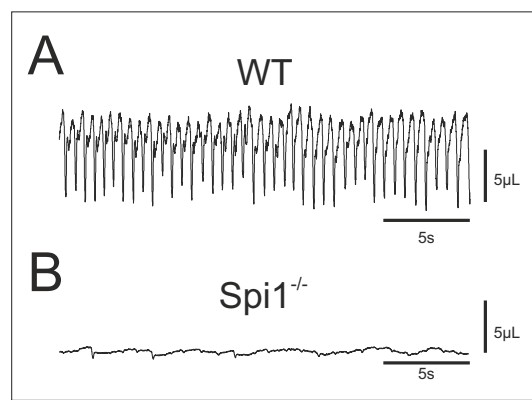

**Figure 7.** Breathing phenotype of *Spi1⁻/⁻* embryos at E18.5. Whole-body plethysmographic recordings of breathing behavior by wild-type (WT) (**A**) and mutant (*Spi1⁻/⁻*) (**B**) embryos at E18.5. Note a complete absence of breathing activity in B despite a detectable low-amplitude signal representing heartbeat activity.

networks of *Spi1*$^{-/-}$ embryos during embryonic development in the absence of microglia are very likely to participate in causing death at birth, with at least one contributing factor being an inability to sustain active breathing movements.

## Discussion

The present study was motivated by the general current lack of knowledge on the role played by microglia in the emergence of neural circuit operation during embryonic development. Microglia are known to govern synapse formation and the maintenance of cell populations comprising central circuits (*Paolicelli et al., 2011*; *Schafer and Stevens, 2015*; *Thion and Garel, 2017*), but how such regulatory processes translate to overall network functioning, and especially that of local circuits underlying rhythmogenic motor behavior, is virtually unknown. Our findings reveal that the complete absence of microglia during *Spi1*$^{-/-}$ mouse embryonic development results in major deficits of the identified neuronal networks responsible for breathing behavior, specifically in relation to their ability to produce an effective central respiratory command.

Respiratory rhythmogenesis relies on an interaction between distinct neuronal groups located in separate regions of the brainstem, which normally become sequentially operational during embryogenesis to produce a stable respiratory-related rhythm from 3 days before birth (at E15.5) onwards (*Thoby-Brisson et al., 2009*). In examining the anatomical and functional characteristics of the two main respiratory neuronal groups (the epF and the preBötC networks) we found that when developing in an environment devoid of microglia, these networks continue to become functional at typical developmental stages, but exhibit anomalies that lead to the generation of an overall slower rhythm and a subsequent inability to sustain breathing behavior at birth. This deficit is likely to arise from a lack of the normal influence of microglia in regulating cell numbers and the proper wiring of the brainstem circuits responsible for respiratory rhythmogenesis during prenatal development. However, we cannot rule out the possibility of indirect effects, especially since the regions in which respiratory networks reside are not those that express a high accumulation of microglia. This therefore raises the possibility that anomalies in the establishment of the central respiratory command in the Spi1 mutant arises from defaults at very early developmental stages (during early migration of progenitors, for example) or in other actors involved brainstem circuit development.

The main abnormality observed in the mutant was a very low respiratory frequency expressed in vitro by isolated brainstem preparations. The centrally generated respiratory rhythm results in part from network interactions between the parafacial respiratory group (epF at embryonic stages) and the preBötC (*Feldman et al., 2013*), both of which are strongly connected to their contralateral counterparts via midline-crossing commissural projections (*Wu et al., 2017*). Thus, the cycle frequency of respiratory activity is determined by (1) the intrinsic rhythm generated by each half-center network, and (2) the strength of the homo- (preBötC/preBötC; epF/epF) and hetero- (preBötC/epF) network synaptic interactions. During embryonic development these dual controlling processes are established between E14.5 and E16.5 (*Thoby-Brisson et al., 2009*), a period during which the epF (intrinsically active at a higher frequency) progressively entrains the intrinsically slower preBötC network, thereby setting the overall respiratory rhythm to an intermediate cycle frequency (*Thoby-Brisson et al., 2009*). Here, we find in the mutant that at the time of their respective emergence, each respiratory network generates rhythmic activity at an abnormally low frequency. In the case of the epF, this lower frequency is likely to be the consequence of a substantially diminished number of constituent neurons. For the preBötC network, anomalies in commissural projections connecting the two bilaterally distributed preBötC neuronal groups, resulting in a diminished inter-network synchronization, are likely to contribute to the slower rhythm generation. Ultimately, when these networks become fully connected at E16.5, their intra- and inter-circuit anomalies result in a reduced frequency of the final respiratory command observed at the level of motor output.

### Premature death of *Spi1*$^{-/-}$ embryos

While *Spi1*$^{-/-}$ mortality at P0 probably results from multiple functional defaults that are not necessarily exclusively CNS related (*Back et al., 2004*), the cause of death is very unlikely related to an immune system deficiency or the development of septicemia that could have arisen due to the absence of microglia. In contrast, peripheral causes are likely to be involved in the premature death of animals

devoid of microglia. Indeed, Spi1 is expressed in hematopoietic stem cells and common lymphoid progenitors impacting not only microglia but also meningeal, choroid plexus, and perivascular macrophages, as well as peripheral immune cells (*Goldmann et al., 2016*). In a recent study, no premature death was observed in transgenic mice in which microglia depletion was specifically restricted to the CNS (*Rojo et al., 2019*), thus directly inferring peripheral deficits, possibly including developmental malformations of the upper airways, lungs and heart, as important contributors to neonatal death of $Spi1^{-/-}$ mutants. Although further investigations are now needed to address such possibilities, our findings nevertheless indicate that, despite the sustained generation of a central, rhythmically organized respiratory motor command, peripheral breathing movement was rapidly undetectable in neonatal mutants. Therefore, major defaults in overall breathing movement control (potentially also at the peripheral level) and/or anomalies in cardiorespiratory activity are very likely contributing significantly to the inability of perinatal Spi1 mutants to survive after birth.

Microglial-related defects in the motor output commands themselves that are produced by hindbrain circuits could be another contributor to mutant immediate postnatal death, especially since in WT embryos, we found microglia to be present with a higher density in brainstem regions where motor nuclei are located. Interestingly, anatomical anomalies were observed in the hindbrain facial nucleus of mutant preparations, although whether these two features are linked and whether such anatomical defaults lead to actual functional deficits remain to be determined. Also, potential anatomical and functional defects in the hypoglossal nucleus, another hindbrain motor nucleus that controls muscles of the upper airways, similarly need to be investigated. Overall, however, a multifactorial combination of deficits at both central and peripheral levels is very probably responsible for the immediate death of newborn pups when embryonic development has occurred in the complete absence of microglia, presumably accelerating the fatal impact of a deficient respiratory command.

## Regional localization and associated roles of microglia in the embryonic brainstem

Microglia are known to enter and invade the CNS early during development (*Mosser et al., 2017*; *Thion and Garel, 2017*; *Angelim et al., 2018*), consistent with an important role in the initial stages of neuronal circuit assembly (*Casano and Peri, 2015*; *Reemst et al., 2016*; *Mosser et al., 2017*). Moreover, microglia exhibit a specific spatiotemporal distribution in the embryonic brain, with a characteristic uneven distribution that coincides with the localization of progenitor niches, loci where neurogenesis and cell division also occur, near developing blood vessels and the formation of axonal tracts (*Ashwell, 1991*; *Fantin et al., 2010*; *Verney et al., 2010*; *Hoshiko et al., 2012*; *Swinnen et al., 2013*; *Squarzoni et al., 2014*). Similarly, in the developing brainstem, we find that microglia also express a preferential spatial distribution, with a higher prevalence not only in hindbrain motor nuclei but also along each side of the hindbrain midline. This latter observation was of particular interest since most brainstem circuits are bilaterally distributed, with their activities coordinated by commissural axonal projections that cross the midline. This is especially relevant for the central networks organizing the respiratory command, where a strict left/right synchronization of motor output is required to produce a coordinated drive to the left and right diaphragm muscles (*Bouvier et al., 2010*).

The commissural connectivity between the central respiratory groups displays redundancy and is extremely complex. For example, a bilateral co-activation of the two epF networks occurs as early as E14.5, with synchronizing signals being conveyed through multiple, but still anatomically undefined commissural pathways (*Thoby-Brisson et al., 2009*). In addition, commissural interneurons, both between the two preBötC networks (*Smith et al., 1991*; *Bouvier et al., 2010*) and between the ventral respiratory groups (VRGs) that contain pre-motoneurons (*Wu et al., 2017*), participate in the synchronization of bilaterally coordinated contractions of inspiratory effector muscles. At the level of the preBötC, this anatomical specificity requires the Slit/Robo signaling pathway (*Bouvier et al., 2010*), but might also involve microglia due to the latter's localization near the midline and their known participation in axonal tract formation elsewhere in the CNS (*Pont-Lezica et al., 2014*; *Squarzoni et al., 2014*). Consistent with this idea, our findings show firstly, that inter-preBötC network projections are considerably reduced in the $Spi1^{-/-}$ embryos, and second, and correspondingly, in slice preparations isolating the preBötC networks from $Spi1^{-/-}$ embryos at E16.5, the normal functional synchronization between the rhythmogenic half-centers is diminished. However, despite this abnormality in the relative timing between left/right preBötC bursts in the mutant, a strict bilateral

synchronization similar to that found in the WT was observed in phrenic (C4) motor nerve recordings from the whole isolated hindbrain at E18.5. This apparent compensation is likely attributable to the redundancy of midline-crossing commissures between the other brainstem centers that are sufficient to offset the weakened local connections between the preBötC half-centers (*Wu et al., 2017*), thereby enabling the coordination between the downstream left and right motor outputs in the $Spi1^{-/-}$ embryo to be maintained.

Without fully disrupting bilateral coupling, the absence of microglia and the resulting reduced number of midline-crossing fibers connecting the two preBötC networks affects other parameters of respiratory pattern generation. Most notably, in the mutant phrenic motor burst output occurs with a significantly slower cycle frequency and more strikingly longer individual bursts than in the WT. On the one hand, the decreased frequency is likely to be the direct consequence of the slower rhythmic command signal emanating from the interacting epF and preBötC networks, both groups separately generating a respiratory-related rhythmic activity at an abnormally slow frequency. On the other hand, the increase in motor burst duration could have arisen from a reduction in the coordination within the preBötC networks and between inspiratory interneurons and VRG pre-motoneurons. This in turn would lead to a less synchronized drive to target motoneurons and thus an overall increase in phrenic motor burst duration in each cycle. Another possibility is that the phrenic motor nucleus itself is affected in the mutant, especially in light of our finding that microglia are normally present in high density in the different brainstem motor nuclei. Thus, when developing in the absence of microglia, the firing properties of these motoneuronal groups, including those in the phrenic nucleus, might be directly affected, thereby contributing to the observed increase in phrenic burst duration.

## Early role of microglia in respiratory network ontogenesis

The essential regulation of neuronal numbers in the construction of neural circuitry during development occurs through several mechanisms in which microglia may participate: neuronal death, survival, and migration (*Thion and Garel, 2017*). In the mouse respiratory system, epF neurons originate in the dB2 domain of rhombomere 5 (r5) then migrate conjointly with facial motoneurons to reach their final location in r6 by E14.5, near the ventral surface of the hindbrain and below the facial motor nucleus (*Dubreuil et al., 2009*; *Ramanantsoa et al., 2011*; *Mellen and Thoby-Brisson, 2012*). Perturbations in epF formation have been detected in different mouse lines genetically modified for transcription factors known to specify epF neuronal components, such as *Phox2b*, *Egr2*, *Lbx1*, and *Atoh1* (*Pagliardini et al., 2008*; *Dubreuil et al., 2009*; *Rose et al., 2009*; *Thoby-Brisson et al., 2009*; *Ruffault et al., 2015*). In most cases, the number of epF neurons was found to be drastically reduced in these mutant animal, with significant resultant effects on epF network function, including both rhythmogenesis and loss of $CO_2$/pH responsiveness. In the $Spi1^{-/-}$, the reduction (by ~31%) of neurons constituting the epF group presumably results from an abnormally high level of epF cell death and/or migratory dysfunction. This diminished epF population size in turn probably explains the production of a slower (by 33%) epF rhythm at E14.5, and indeed, also that of fictive respiration observed from E16.5 onwards in isolated hindbrain preparations, where rhythmic burst activity in the phrenic motor roots arises from both the epF and the now active preBötC generators operating in combination. Since at this later embryonic stage the epF is known to entrain the preBötC (*Thoby-Brisson et al., 2009*), a slower intrinsic rhythm generated by the epF in the mutant will drive the preBötC at an equally reduced frequency (but note, the preBötC itself generates a slower rhythm; see below), with a resultant production of slower phrenic motor bursting. In contrast, the epF's intrinsic sensitivity to pH changes at E14.5 is preserved in the mutant, indicating that despite a reduction in constituent neurons, the normal chemosensing capability of this neuronal group is preserved. This functional property also persists later in development, since acidifying the aCSF bathing isolated brainstem preparations at E18.5 still evoked a frequency increase of respiratory activity recorded at the level of phrenic roots. Together these findings therefore indicate that although the complete neuronal composition of epF circuitry early during embryonic development depends on the presence of microglia, nevertheless, in the latter's absence the network's basic rhythmogenic and chemosensitivity properties remain.

The preBötC network originates from the V0v domain in r6/7, and is composed of a heterogeneous population of Dbx1-specified interneurons that migrate radially from their territory of origin (near the *sulcus limitans* bordering the fourth ventricle) to their final position in the ventro-lateral area in the vicinity of the nucleus ambiguus (*Bouvier et al., 2010*; *Gray et al., 2010*; *Mellen and Thoby-Brisson,*

*2012*). Potential anatomical deficits in the preBötC network population are more complicated to detect in general as it does not express any specific molecular characteristics enabling anatomical marker detection. However, our physiological results indicate that this network in the mutant is sufficiently preserved, if affected at all, in its ability to produce inspiration-related rhythmic activity, albeit at a slower frequency. This frequency reduction could result from a diminished connectivity between inspiratory interneurons as indicated by the reduced number of commissural projections between the preBötC half-centers (as discussed above) and a consequent less efficient synchronization between the two preBötC groups. Another plausible explanation resides with the known role of microglia in shaping and refining synapses during development (*Thion and Garel, 2017*; *Thion et al., 2018*), in their formation and maturation (*Miyamoto et al., 2016*), and in regulating synaptic activity (*Ben Achour et al., 2010*; *Li et al., 2012*; *Pascual et al., 2012*). Such multiple influences might also contribute to the establishment of the preBötC network, in which appropriate interneuronal connections and activity levels are required for proper respiratory rhythmogenesis. Of further relevance is the fact that Brain-Derived Neurotrophic Factor (BDNF) is known to modulate preBötC activity through an activation of Trkb receptors expressed by inspiratory neurons and is required for the normal development of the central respiratory rhythm (*Erickson et al., 1996*; *Balkowiec and Katz, 1998*; *Thoby-Brisson et al., 2003*; *Bouvier et al., 2008*). Indeed, mice lacking BDNF display severe deficits in breathing control, including depressed ventilation and an irregular respiratory rhythm. On the other hand, microglia are able to release BDNF that can cause a disinhibition of neural circuits and increase network excitability (*Coull et al., 2005*; *Trang et al., 2011*; *Ferrini et al., 2013*). A further possibility, therefore, is that microglia also serve as a source of BDNF for the preBötC network, and whose absence during embryonic development leads to abnormal preBötC function. Although beyond the scope of the present study, this hypothesis clearly warrants further investigation.

The roles of microglia in neurogenesis, axonal tract formation, and proper circuit wiring are evidently relevant to the developmental establishment of a functional central respiratory command, as evidenced here by our *Spi1$^{-/-}$* analysis. Nonetheless, despite anatomical and functional disturbances in respiratory network emergence, the absence of microglia during embryonic development does not totally prevent the formation of neural circuits capable of fundamental respiratory-related rhythmogenesis. This is probably due to additional strong and even redundant mechanisms that contribute to establishing the integral central command for such a vital function.

## Microglia in relation to gender and age

The number and morphology of microglia throughout brain development are dependent upon the sex and age of an individual, as well as the particular brain region (*Schwarz et al., 2012*). So far, however, sex-specific differences in such parameters have only been found at postnatal stages, whereas in the prenatal period, brain microglia numbers and morphologies are apparently not affected by animal gender (*Schwarz and Bilbo, 2012*). Accordingly, we did not discriminate between male and female embryos in our experiments, although we have no evidence to suggest that the dysfunctions of the central respiratory command observed in the *Spi1$^{-/-}$* embryo was other than a consequence of the absence of microglia, whatever the sex of the embryo.

## Concluding remarks

Microglia colonize the CNS early during embryogenesis and contribute to various developmental processes. The early roles of embryonic microglia have only begun to be described in rodents, including the regulation of axonal tract formation (*Pont-Lezica et al., 2014*; *Squarzoni et al., 2014*), positioning of migratory interneurons (*Squarzoni et al., 2014*), modulation of neurogenesis and synapse refinement (for review see *Witcher et al., 2021*). Thus, microglia are implicated in a wide range of processes, rendering these cells important contributors to the assembly and refinement of central neuronal circuits during ontogenesis. In the context of breathing control, the possible roles of microglia have so far been mainly investigated in relation to inflammatory responsiveness and its impact on rhythm generation, autoresuscitation, chemoreception, and plasticity at postnatal ages (*Huxtable et al., 2011*; *Huxtable et al., 2013*; *Tadmouri et al., 2014*; *Lorea-Hernández et al., 2016*; *Camacho-Hernández et al., 2019*; *Beyeler et al., 2020*). Here, in a first description of the distribution and role of microglia in the embryonic hindbrain, we provide new data on the contribution of microglia to the early anatomical and functional development of brainstem networks responsible

for respiratory rhythmogenesis. Our findings thus underline the critical role played by these brain-resident macrophages in the ontogeny of central neuronal circuits, including those controlling vital physiological functions. Future work now requires identifying the cellular actors, signaling pathways, and mechanistic processes supporting this role.

## Materials and methods

All procedures employed in this study were conducted in accordance with the local animal welfare ethics of the University of Bordeaux as well as national and European committee regulations. Experiments were performed on E14.5 to E18.5 mouse embryos of either sex obtained from pregnant females raised in our laboratory's breeding facility. All efforts were made to minimize animal suffering and to reduce the number of animals used, in accordance with the European Communities Council Directive (2010/63/UE).

### Animals

The Spi1 mouse strain, exhibiting a complete absence of microglial cells in the nervous system (*Back et al., 2004*), was maintained and bred in our animal housing facility. Because homozygous animals die rapidly after birth, heterozygous males and females were intercrossed to generate litters containing $Spi1^{-/-}$ mutant embryos, which constituted less than 25% of each litter. WT ($Spi1^{+/+}$) control animals were littermate embryos. All experiments were performed blind on embryos between E14.5 and E18.5 and the genotype of each embryo was established a posteriori. All heterozygous embryos were discarded from further analysis. To enable experiments on precisely dated embryos, male and female mice were placed in the same cage for a single night. In the event of mating, a vaginal plug was detected the following morning, which was then considered to be E0.5.

In another set of experiments, in order to determine the distribution of microglial cells in the control hindbrain, we used the $Cx3cr-1^{GFP}$ knock-in/knock-out mouse line that expresses enhanced green fluorescent protein (EGFP) in brain microglia (*Jung et al., 2000*). E18.5 transgenic embryos were obtained by mating $Cx3cr-1^{GFP}$ males (initially purchased from Jackson Laboratory) with C57Bl/6J mice.

### Plethysmographic recordings

Breathing behavior was assessed for un-anaesthetized and unrestrained embryos at E18.5 using a whole-body plethysmograph. After a C-section of a pregnant mouse at E18.5, embryos were harvested, placed under a heating lamp and gently mechanically stimulated until they started breathing autonomously. Then embryos were placed inside a 50-ml chamber for recording sessions lasting 5 min. The chamber was also placed under a heating lamp to avoid cooling of experimental animals and to maintain a constant temperature throughout the entire recording period. The chamber was connected to a data acquisition system (Emka Technologies, France) capable of measuring flow and pressure changes within the chamber using Iox2 software. Data were sampled at 1 kHz and recordings were analyzed offline using PClamp10 software (Molecular Devices, USA). Breathing frequency was calculated on a breath-to-breath basis. Because of the small size of animals at E18.5 and their very limited respiratory volumes, to avoid erroneous data acquisition, other conventionally tested breathing parameters such as tidal volume, ventilation, and the duration of inspiration and expiration, were not measured.

### In vitro preparations

Pregnant mice were killed by cervical dislocation when their embryos had reached either E14.5, E16.5, or E18.5. Embryos were excised from the uterine corns and their individual uterine bags, and prior to experimental use for those at E14.5 and E16.5, were placed in aCSF that was continuously supplemented with oxygen at a temperature not exceeding 24°C. The aCSF solution was composed of (in mM): 120 NaCl, 8 KCl, 1.26 $CaCl_2$, 1.5 $MgCl_2$, 21 $NaHCO_3$, 0.58 $NaH_2PO_4$, 30 glucose, and at pH 7.4. When needed, the pH of the aCSF was lowered to 7.2 by decreasing the $NaHCO_3$ concentration to 10.5 mM and adjusting NaCl to 130.5 mM.

Two types of in vitro preparations at various stages of isolation were used in the present study: at E14.5 or E18.5, isolated medullary (brainstem) preparations that contain the two main respiratory networks, the epF and the preBötC; and at E16.5, transverse brainstem slices containing the preBötC

network alone. All preparations were dissected in aCSF maintained at 4°C and continuously bubbled with oxygen. Embryos were first decerebrated then the brainstem was isolated from the embryos' head by a rostral section performed at the junction between the rhombencephalon and mesencephalon, and a caudal section made below the sixth cervical roots. Surrounding tissue was carefully removed in order to keep intact ventral cervical roots required for subsequent extracellular axonal recordings. In addition for calcium imaging experiments, the pia was carefully removed on brainstem preparations at E14.5 to allow dye penetration and cellular imaging. Transverse brainstem slices were further obtained by serially sectioning in the rostral to caudal direction isolated hindbrains mounted in a low melting point agar block using a vibratome (Leica VS 1000, Leica Microsystems, France). A 450-μm-thick slice isolating the preBötC was obtained at an axial level that was ~300 μm more caudal to the posterior limit of the facial nucleus. Other anatomical landmarks were also used to identify the appropriate tissue region, such as the opened fourth ventricle, the inferior olive, nucleus ambiguus, and hypoglossal nucleus that should all be detectable in the required slice (*Thoby-Brisson et al., 2005*; *Ruangkittisakul et al., 2011*). All in vitro preparations were then positioned ventral (brainstem preparations) or rostral (transverse slice) side up in the recording chamber and continuously bathed in circulating oxygenated aCSF at 30°C. To allow the preparations to recover from the slicing procedure and permit subsequent calcium imaging (see below), a period of 20–30 min was respected before any recording sessions commenced.

## Electrophysiological recordings of neuronal activity

In brainstem preparations at E18.5, activity of the phrenic nerve ventral roots (C4) was recorded using a suction electrode fabricated from glass tubing (Harvard Apparatus, Germany) broken at the tip to match the diameter of the monitored C4 root. In transverse brainstem slices at E16.5, preBötC population activity was recorded using one or two glass macropipettes (tip diameter 80–100 μm) positioned on the surface of the slice at the level of one or two bilaterally distributed regions containing the preBötC respiratory networks (in the proximity of the nucleus ambiguus). The recording pipettes were filled with aCSF and connected through a silver wire to a high-gain amplifier (AM Systems, USA). Signals were filtered (bandwidth 3 Hz to 3 kHz), rectified and integrated (to provide integrated traces; time constant 100ms; Neurolog, Digitimer, England), recorded and analyzed offline on a computer through a Digidata 1440 interface and PClamp10 software (Molecular Devices, USA). The frequency of spontaneous respiratory-related rhythmic activity, its coefficient of variation (SD/mean) and burst durations were measured over periods of 2–3 min. Statistical differences were estimated using Student's *t*-test or the Mann–Whitney Rank Sum test where appropriate, with differences considered significant at $p < 0.05$.

## Calcium imaging

The activity of multiple epF neurons at E14.5 was monitored simultaneously using calcium imaging procedures fully described elsewhere (*Thoby-Brisson et al., 2009*). Briefly, isolated brainstem preparations with the pia carefully removed were first incubated in the dark for 45 min at room temperature (RT) in a solution of oxygenated aCSF containing the cell-permeable calcium indicator dye Calcium Green-1 AM (10 μM; Life Technologies, France). After dye-loading, preparations were positioned ventral side up in the recording chamber. Before subsequent image acquisition, a 30-min delay was observed to wash out excess dye and enable the preparation to stabilize in oxygenated aCSF at 30°C. Fluorescent signals were captured through a FN1 upright microscope (Nikon, Japan) equipped with an epifluorescent illumination system and a fluorescein filter coupled to an Andor camera (Oxford Instruments, Belfast, UK). Images (200-ms exposure time) were acquired over periods lasting 120 s and analyzed using customized software kindly provided by Dr N. Mellen (*Mellen and Tuong, 2009*).

## Pharmacological treatment

SP obtained from Sigma (Merck, St Louis, USA) was dissolved in aCSF at 0.5 μM and bath applied for 10 min. Cycle frequency measurements of respiratory-related rhythmic activity were made during the last 3 min of drug application. Frequency values are given as means ± standard error of the mean, and statistical significance was tested using a Student's *t*-test. Differences were considered to be statistically significant at $p < 0.05$.

## Immunostaining

For immunostaining procedures, brainstem preparations were placed in 4% paraformaldehyde in phosphate-buffered saline (PBS) 0.1 M for 2–3 hr for tissue fixation. Transverse or sagittal frozen sections were obtained by placing brainstems overnight in a 20% sucrose-PBS solution for cryoprotection, then embedding them in a block of Tissue Tek (Leica Microsystems, France) and sectioned at 30 µm using a cryostat (Leica Microsystems, France). To limit nonspecific labeling and favor tissue penetration of antibodies, preparations (slice or brainstem) were incubated for 90 min in a solution of PBS containing 0.3% Triton X-100 and 1% BSA. Primary antibodies were then applied overnight at RT and under slight agitation. As primary antibodies, we used a goat anti-ChAT (1/100; Merck-Millipore, France) or a mouse anti-islet 1,2 (1/250; DSHB, USA) as markers for cholinergic motoneurons, a rabbit anti-Iba1 (1/500; Wako, Germany) as a marker for microglia, and a rabbit anti-NK1R (1/10,000; Merck-Sigma, France) and a mouse anti-Phox2b (1/100; Santa Cruz Biotechnology, Germany) as markers for NK1R- and Phox2b-expressing neurons, respectively. In addition, for experiments aiming to label commissural projections connecting the two bilaterally positioned preBötC networks, an incision was made in the tissue where preBötC neuron axons exit the network and a crystal of Alexa Fluor 488 dextran dye was carefully applied at this location, then the crystal dye was left to migrate for at least 4 hr before the slice preparation was fixed overnight. To amplify primary antibody signals, after several washes, preparations were incubated with secondary antibodies for 90 min at RT. These secondary antibodies (1/500; Thermo Fisher, France) included an Alexa Fluor 647 or Alexa Fluor 568 donkey anti-rabbit antibody, an Alexa Fluor 488 anti-mouse antibody, an Alexa Fluor 568, and an Alexa Fluor 647 donkey anti-goat antibodies. Immunostained slices and isolated brainstem preparations were mounted in Vectashield Hard Set medium (Eurobio, France) cover-slipped and kept in the dark until imaging was conducted using an epifluorescence microscope (Olympus, Japan) or a Zeiss confocal microscope (LSM 900, Zeiss, France).

## CLARITY-based treatment of embryonic CX$_3$CR-1$^{GFP}$ hindbrain tissue

Microglial cells were located in whole hindbrains of CX$_3$CR-1$^{GFP}$ transgenic mice embryos (E18.5) that had an EGFP sequence replacing the first 390 bp of the coding exon (exon 2) of the chemokine (C-X3-C motif) receptor 1 (Cx3cr1) gene (*Jung et al., 2000*). Hindbrain tissues were cleared with hydrogel embedding derived from CLARITY methodology. Brainstems were dissected as detailed above, with the pia carefully removed. All steps of the protocol were conducted using 12 ml Falcon tubes wrapped in aluminum foil to preserve fluorescence. Brainstems were incubated at 4°C, under gentle agitation for 24–36 hr in a fixative mixture of 4% acrylamide (Merck), 4% paraformaldehyde and 0.25% VA-044 thermal initiator (Fujifilm, Wako, Germany) in PBS 0.1 M. Samples were subsequently incubated for at least 6 hr in a solution of 4% acrylamide with VA-044 in PBS 0.1 M, in a shaking incubator at 37°C in order to allow polymerization of the hydrogel. Because oxygen inhibits polymerization, the solution was isolated in each tube with a layer of peanut oil and the tube was sealed with Parafilm. Brainstem preparations were then rinsed thoroughly at RT under agitation, in three to four baths of PBS-azide (PBS 0.1 M with 0.01% sodium azide) in order to remove excess gel from the tissue.

For passive clearing, hydrogel-embedded samples were then incubated at 37°C under agitation in a detergent clearing solution (4% sodium dodecyl sulfate in 200 mM boric acid, pH 8.5 with 0.4% lithium hydroxide monohydrate) for 2 weeks. This solution was replaced every 3–4 days. The cleared blocks were then washed thoroughly in PBS-azide-Triton X-100 (PBS 0.1 M with 0.01%sodium azide and 1% Triton X-100) at RT for 48 hr with the solution being replaced twice daily.

For immunostaining, cleared samples were rinsed in three successive baths of 0.01 M PBS at RT under shaking. Then, they were incubated for 7 days, at 37°C under agitation in 0.1 M PBS containing 6% bovine serum albumin, 1% Triton X-100, 0.01% sodium azide and primary antibodies: a goat anti-choline acetyl transferase (ChAT, 1/50; Millipore) and chicken anti-GFP (1/50; Aves). Afterwards, they were washed at RT under shaking, in seven baths of PBS 0.01 M (each for 1 hr), then they were incubated at 37°C under agitation for another 7 days cycle in secondary antibodies: a donkey anti-chicken IgG conjugated to Alexa Fluor 488 and a donkey anti-goat IgG conjugated to Alexa Fluor 647 (both from Invitrogen, France) diluted 1/200 in the same buffer as the primary antibodies. Samples were washed again at RT under shaking in seven baths of PBS 0.01 M (each for 1 hr). In a final step, cleared/immunostained blocks were incubated in a refractive index matching solution consisting of 0.1 M PBS with 75% diatrizoic acid, 70% D-sorbitol and 23% N-methyl-D-glutamine for 2 days under shaking at

RT. For confocal microscope observation (see below), the samples were mounted in the same solution on glass slides equipped with Coverwell Imaging Chambers (Electron Microscopy Sciences, USA).

## 3D image acquisition and processing

Clarified embryonic hindbrain samples (2 mm thick) were imaged using a Zeiss confocal laser scanning microscope (Zeiss LSM900) equipped with a ×10 dry objective (Zeiss, EC Plan-Neofluar, numerical aperture = 0.3, working distance = 5.2 mm). The imaging volume was 6 mm × 7 mm × 1.8 mm with a voxel size of 2.5 µm × 2.5 µm × 5 µm. Zeiss Zen software was used to acquire and fused z-stacks of multichannel tile images (each tile image size was 512 × 512 pixel). ImageJ (Fiji) was used to process and analyze volume images for microglia.

With the cleared whole-hindbrain tissue, quantification of the distribution of microglia (GFP⁺) was performed by counting cells in the 3D volume. Stacked images were processed for enhancement (subtracted background, median 3D filter), and a 3D suite plugin was used to segment microglia while a 3D density map was generated with the 3D density map plugin (*Ollion et al., 2013*). Finally, a lookup table was applied to the resultant image in order to link cell density values to different colors.

## Acknowledgements

This work was supported by an 'Equipe FRM' funding (DEQ20170336764) to MTB and IRME funding to OP. We thank Anne Fayoux for her contribution to the maintenance of mouse lines and the handling of dated mating.

## Additional information

### Competing interests

Muriel Thoby-Brisson: Reviewing editor, *eLife*. The other authors declare that no competing interests exist.

### Funding

| Funder | Grant reference number | Author |
|---|---|---|
| Fondation pour la Recherche Médicale | DEQ20170336764 | Muriel Thoby-Brisson |
| Institut pour la Recherche sur la Moelle épinière et l'Encéphale | | Olivier Pascual |

The funders had no role in study design, data collection, and interpretation, or the decision to submit the work for publication.

### Author contributions

Marie-Jeanne Cabirol, Formal analysis, Investigation, Methodology; Laura Cardoit, Investigation, Methodology; Gilles Courtand, Data curation, Formal analysis, Methodology; Marie-Eve Mayeur, Data curation; John Simmers, Conceptualization, Writing - review and editing; Olivier Pascual, Conceptualization, Data curation, Funding acquisition, Writing - original draft, Writing - review and editing; Muriel Thoby-Brisson, Conceptualization, Data curation, Formal analysis, Supervision, Funding acquisition, Validation, Investigation, Methodology, Writing - original draft, Project administration, Writing - review and editing

### Author ORCIDs

John Simmers (iD) http://orcid.org/0000-0002-7487-4638
Muriel Thoby-Brisson (iD) http://orcid.org/0000-0003-3214-1724

### Ethics

All efforts were made to minimize animal suffering and to reduce the number of animals used, in accordance with the European Communities Council Directive (2010/63/UE).

Decision letter and Author response
Decision letter https://doi.org/10.7554/eLife.80352.sa1
Author response https://doi.org/10.7554/eLife.80352.sa2

## Additional files

### Supplementary files
• MDAR checklist

### Data availability
All data generated or analysed during this study are included in the manuscript.

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
