## [Editor Report]

This study presents fundamental experimental data from a mutant mouse model lacking microglia (Pu.1-/- mouse line) indicating that these cells have an important role in the embryonic establishment of critical neural circuits in the brainstem generating breathing motor behavior in mice. The authors examined in comparison to wild-type animals the anatomical and functional characteristics of two main respiratory neuronal groups-in the embryonic parafacial (epF) and the preBötzinger complex (preBötC) respiratory regions that operate together in the developing brainstem to generate the rhythmic neural activity necessary to establish normal breathing behavior and ensure survival at birth. Convincing experimental evidence is presented indicating that these respiratory networks become functional at typical developmental stages in the absence of microglia but exhibit anomalies in endogenous rhythm generation (slower respiratory rhythm). The authors' data suggest that these deficits are associated with reduced cell numbers of active neurons and abnormal rhythmogenesis in epF and reduced commissural axonal projections affecting bilateral activity synchronization of the preBötC circuits generating inspiratory rhythm.

---

## [Decision Letter]

**Decision letter after peer review:**

Thank you for submitting your article "Microglia shape the embryonic development of mammalian respiratory networks" for consideration by *eLife*. Your article has been reviewed by 3 peer reviewers, including Jeffrey C Smith as Reviewing Editor and Reviewer #1, and the evaluation has been overseen by Ronald Calabrese as the Senior Editor. The following individual involved in the review of your submission has agreed to reveal their identity: Eric Herlenius (Reviewer #3).

Essential revisions:

1) Since Pu.1 is not specific to microglia and is expressed in hematopoietic stem cells and common lymphoid progenitors impacting meningeal, choroid plexus, and perivascular macrophages, the authors need to provide additional caveats about neonatal death since in another specific depletion model of microglia, for example, the FIRE mice (Rojo et al., 2019), there are no reported neonatal deaths. Have the authors detected other, including peripheral, deficits of the Pu.1 -/- mice that may have contributed to their immediate postnatal death?

2) The authors should provide where available, or at least discuss the need for, some quantitative information on the kinetics of microglia colonization of the different brainstem nuclei to present a more precise picture of the observed accumulations from E14.5 to E18.5.

3) In Figure 3, the size of the VII and epF nuclei appear to be a bit smaller in the mutant conditions. Did the authors carefully analyze this? Similarly, a quantification of the number of Phox2b-/+, Islet1,2-/+ could be performed in control and mutant as it appears different at least in Figure 3B. If there are no effects, the authors should state this.

4) The authors should be more cautious when referring to microglia using Iba1 or GFP staining in the cx3cr1gfp mouse line as both stain microglia, meningeal, choroid plexus, and perivascular macrophages contrary to P2ry12 that specifically labels microglia in the brain parenchyma. Please add a sentence about the specificity of the staining.

5) The authors should more clearly discuss the fact that the epF and preBotC regions associated with the perturbations of rhythm exhibit few or no microglia during embryonic development, and thus direct localized effects on these neurons due to microglia deletion would be unlikely.

*Reviewer #1 (Recommendations for the authors):*

After going through this manuscript several times, I cannot find major issues that require revision of the present form of the layout of the scientific rationale behind the experimental design, or the technical execution of the experiments. I think that the appropriate caveats about the detailed microglial-related mechanisms, while not investigated, have been covered in the Discussion.

*Reviewer #2 (Recommendations for the authors):*

I believe that the authors should address the following concerns before the manuscript is suitable for publication in *eLife*.

– Regarding the fact that the Pu.1 is not specific to microglia, the authors downtown their conclusion about neonatal death although microglia certainly partially participate in this phenotype explain that in other specific depletion models of microglia, there is no effect on neonatal cell death.

– The kinetics of microglia colonization of the different nuclei in the brainstem should be quantified to have a precise idea of the observed accumulations from E14.5 to E18.5.

– In Figure 3, the size of the VII and epF nuclei appear to be a bit smaller in mutant conditions. Did the authors carefully analyze this? Similarly, a quantification of the number of Phox2b-/+, Islet1,2-/+ could be performed in control and mutant as it appears different at least in Figure 3B. If there are no effects, the authors should state it.

– The authors could be more cautious when referring to microglia using Iba1 or GFP staining in the cx3cr1gfp mouse line as both stain microglia, meningeal, choroid plexus, and perivascular macrophages contrary to P2ry12 that specifically labels microglia in the brain parenchyma. Adding a sentence about the specificity of the staining should be enough.

– The authors report a nice effect on the epF and preBotC neurons but do not clearly discuss the fact that some of the observed effects are in regions that exhibit few or no microglia during embryonic development.

– In Figure 3C, legends should clearly state that the number of quantified cells is Phox2b+, Islet1,2-. In addition, the annotation on the y-axis "Nb of cells/epF" is unclear as it is cell number and not cell density.

– The authors could discuss the fact that maybe the lower number of epF cells could lead to a decreased firing associated with decreased rhythm.

*Reviewer #3 (Recommendations for the authors):*

Excellent paper, nice hypothesis, and experiments.

Immediate death was convincingly shown to be, at least partly due to inadequate, but still existing respiration-related neural networks neurons and commensurable connection. However, other pathologies?

The manuscript lacks a description of other deficits of the Pu.1 -/- mice, that might have contributed to their early immediate postnatal death. Perhaps this is already published, although hard to find.

Grateful for referral to relevant work or basic data on mouse postnatal phenotype. Eg Heart formation, upper airways, and lungs – any obvious macro or microscopical deficits.

Other deficits are likely to participate in causing death at birth, with at least one contributing factor being an inability to sustain active breathing movements.

If not please state and include it in the discussion.

Row 356-357; "While Pu.1-/- mortality at P0 probably results from multiple functional defaults that are not necessarily exclusively CNS-related (Back et al., 2004) –

This paper, focused on erythrocytes, the phenotype described is that the mice are anemic at birth. Is more extensive phenotyping in other papers available?

Please elaborate in the discussion.

In my opinion good that you mentioned "the cause of death is very unlikely related to an immune system deficiency or the development of septicemia that could have arisen due to the absence of microglia.." BUT Heart Upper airways and Lung development normal?

Reduced but existing frequency of CPG- Lung development and fetal functional breathing exist – inside the womb – Ultrasound? This is not necessary but might indicate if breathing lung movement exists before birth or not but not enough to enable postnatal life.

– '362-364; Therefore, major defaults in overall breathing movement control (potentially also at the peripheral level) and/or anomalies in cardio-respiratory activity are very likely to be major contributors to the inability of perinatal Pu.1 mutants to survive.

Agree, but are there ANY signs of Peripheral deficits; eg upper airway, heart, or lung anatomical /functional deficits?

Thus, in summary, excellent paper but please add some more refs/discussion or data to underline that the deficits that you have shown are a major contributor to immediate postnatal death.

ANY sign of Peripheral deficits; eg upper airway, heart, or lung anatomical /functional deficits?

---

## [Author Response]

Reviewer #1 (Recommendations for the authors):After going through this manuscript several times, I cannot find major issues that require revision of the present form of the layout of the scientific rationale behind the experimental design, or the technical execution of the experiments. I think that the appropriate caveats about the detailed microglial-related mechanisms, while not investigated, have been covered in the Discussion.

We are grateful for this highly positive appreciation of our work.

Reviewer #2 (Recommendations for the authors):I believe that the authors should address the following concerns before the manuscript is suitable for publication in eLife.– Regarding the fact that the Pu.1 is not specific to microglia, the authors downtown their conclusion about neonatal death although microglia certainly partially participate in this phenotype explain that in other specific depletion models of microglia, there is no effect on neonatal cell death.

We thank the referee for having drawn our attention to the Rojo et al., (2019) article. This paper indeed indicates that the absence of microglia selectively in the brain does not affect the survival of newborn mouse pups, in contrast to what we observed in the Pu.1 mutants. This therefore argues in favor of a multiple causes of death in the latter mutant in which peripheral deficits potentially affecting meningeal, choroid plexus and perivascular macrophages maybe associated with the central nervous defaults found in our study. We now address this specific point in a new paragraph of the Discussion (pages 13-14), and to also respond to related comments of reviewer 3 (see below).

Note that we started our experiments using the Pu.1 mutant mouse line in 2014, a long time before the FIRE mouse line was established. It was therefore not possible for us to reproduce all our experiments in parallel with this new mouse line, although we definitively should include mention of the Rojo et al., data in our Discussion. Importantly, these latter findings also provide an explanation for neonatal death in the Pu.1 mutant while the central respiratory networks, despite being significantly affected and unable to sustain actual breathing, still remain capable of generating patterned rhythmic activity.

– The kinetics of microglia colonization of the different nuclei in the brainstem should be quantified to have a precise idea of the observed accumulations from E14.5 to E18.5.

We agree with the referee that the developmental kinetics of brainstem tissue invasion by microglia is not described in detail in our study. However, our goal was not to address this particular issue, but rather to show that at key prenatal developmental stages of the respiratory neuronal network (E14.5 for the epF and E16.5 for the preBötzinger complex) microglia are indeed present in brainstem areas hosting the respiratory neuronal networks. We therefore feel that presenting images of microglia colonization at these 3 specific points of development provides a sufficiently supportive basis for our study, and we hope that the reviewer accepts that establishing the detailed developmental dynamic of brainstem microglia colonization is beyond the scope of our study.

– In Figure 3, the size of the VII and epF nuclei appear to be a bit smaller in mutant conditions. Did the authors carefully analyze this? Similarly, a quantification of the number of Phox2b-/+, Islet1,2-/+ could be performed in control and mutant as it appears different at least in Figure 3B. If there are no effects, the authors should state it.

This is actually an excellent observation made by the reviewer, which clearly we missed. This led us to perform a new analysis by measuring the rostro-caudal extension of the facial nucleus in several preparations (n = 11) of each genotype. This led to the confirmation that the size of the facial nuclei was indeed significantly shorter in mutant preparations, and we have added a panel to Figure 3 and corresponding text (page 6) to present this new result. We have also extended the Discussion with an additional paragraph addressing a possible link between the preferential distribution of microglia around hindbrain motor nuclei (as described in Figure 1) and this new observation of a smaller facial nucleus in the mutant, with possible associated functional deficits (page 14).

We are unsure as to what the reviewer means by Phox2b-/+, Islet1,2-/+ cells? We guess that the reviewer refers to epF cells, which are defined as Phox2b+/Islet1,2- cells. We had in fact counted these as quantified in the graph of Figure 3, panel C (see also corresponding text). However the original legend was probably not sufficiently clear about the cellular phenotype of the epF cells counted, and we have modified the legend accordingly.

– The authors could be more cautious when referring to microglia using Iba1 or GFP staining in the cx3cr1gfp mouse line as both stain microglia, meningeal, choroid plexus, and perivascular macrophages contrary to P2ry12 that specifically labels microglia in the brain parenchyma. Adding a sentence about the specificity of the staining should be enough.

The reviewer is correct in pointing out that P2ry12 might be a more specific staining for microglia than the markers employed in our study. Unfortunately, the anti-P2ry12 antibody from Anaspec is not available anymore and the others disposable ones are not specific enough in our hands. But, in the isolated brainstem preparations we used (whole brainstems or slices), the pia and the choroid plexus were removed before staining and clarity procedures were applied. Thus neither meningeal nor choroid plexus macrophages were included in our preparations, and so we are confident that potential spurious positive signals arising from perivascular macrophages would have been minimal. We have added a sentence in the text to emphasis this point (page 5).

– The authors report a nice effect on the epF and preBotC neurons but do not clearly discuss the fact that some of the observed effects are in regions that exhibit few or no microglia during embryonic development.

This is an excellent point that we now address in the Discussion (page 12).

– In Figure 3C, legends should clearly state that the number of quantified cells is Phox2b+, Islet1,2-. In addition, the annotation on the y-axis "Nb of cells/epF" is unclear as it is cell number and not cell density.

Changes have been made accordingly to the figure and its corresponding legend. The y axis does in fact indicate the total number of epF cells but not cell density.

– The authors could discuss the fact that maybe the lower number of epF cells could lead to a decreased firing associated with decreased rhythm.

As stated above, we apologize for not having treated this point more clearly in the Discussion, especially for readers not overly familiar with the respiratory field. The text has now been modified, hopefully to be more explicit (pages 12-13).

Reviewer #3 (Recommendations for the authors):Excellent paper, nice hypothesis, and experiments.

Thank you!

Immediate death was convincingly shown to be, at least partly due to inadequate, but still existing respiration-related neural networks neurons and commensurable connection. However, other pathologies?The manuscript lacks a description of other deficits of the Pu.1 -/- mice, that might have contributed to their early immediate postnatal death. Perhaps this is already published, although hard to find.Grateful for referral to relevant work or basic data on mouse postnatal phenotype. Eg Heart formation, upper airways, and lungs – any obvious macro or microscopical deficits.Other deficits are likely to participate in causing death at birth, with at least one contributing factor being an inability to sustain active breathing movements.If not please state and include it in the discussion.Row 356-357; "While Pu.1-/- mortality at P0 probably results from multiple functional defaults that are not necessarily exclusively CNS-related (Back et al., 2004) –This paper, focused on erythrocytes, the phenotype described is that the mice are anemic at birth. Is more extensive phenotyping in other papers available?Please elaborate in the discussion.In my opinion good that you mentioned "the cause of death is very unlikely related to an immune system deficiency or the development of septicemia that could have arisen due to the absence of microglia.." BUT Heart Upper airways and Lung development normal?Reduced but existing frequency of CPG- Lung development and fetal functional breathing exist – inside the womb – Ultrasound? This is not necessary but might indicate if breathing lung movement exists before birth or not but not enough to enable postnatal life.– '362-364; Therefore, major defaults in overall breathing movement control (potentially also at the peripheral level) and/or anomalies in cardio-respiratory activity are very likely to be major contributors to the inability of perinatal Pu.1 mutants to survive.Agree, but are there ANY signs of Peripheral deficits; eg upper airway, heart, or lung anatomical /functional deficits?

In global response to all the above comments, which are essentially addressing the same concern: in regard to the lack of a description in the Pu.1 mutant of signs of potential anatomical deficits or abnormal organ development, including at the peripheral level, we confess that we did not conduct any specific investigations on this aspect. Our major goal was to decipher the role of microglia in the developmental establishment of the central respiratory command, rather than to identify the cause(s) of death in newborn mice devoid of microglia. And indeed, as indicated by this reviewer, the available literature is extremely poor concerning the gross and fine anatomy of Pu.1 mutant mice. In the McKercher (1996) paper (related to Scott et al., 1994) where the original Pu.1 mutant line was first used, it is stated that the mutant’s appearance was ‘unremarkable’, except for individuals occasionally exhibiting a slightly darker red liver, suggesting potential hepatic functional deficits. In addition to anomalies in the hematopoietic lineage described in this paper, the main cause of death that occurs a few days after birth was proposed to be linked to septicemia. In the Back et al., (2004) study, another Pu.1 mutant line generated (which dies at embryonic stages) exhibited anomalies in the liver and in erythropoiesis. Furthermore, Beers et al., (2006) observed a lack of macrophages, neutrophils, T and B cells and CNS microglia. But in none of these publications could we find a description of gross anatomy anomalies or any organ deficits in Pu.1 mutants. Thus, at the present time, a comprehensive anatomical and functional characterization of Pu.1 mutants is lacking.

We should also add that in our experiments, a small proportion of mutant embryos seemed to die early during embryonic development (around e15.5), with entire embryo’s body turning white in appearance with extremely soft tissue. These subjects were obviously discarded from any subsequent analysis.

Nevertheless, due to the multiple roles of microglia in developmental processes, it is highly likely that brainstem circuits are not the only affected structures in the Pu.1 mutant. Indeed, reviewer 2 pointed to a study (Rojo et al., 2019) in which the authors found that the specific depletion of microglia in the brain during development did not lead to premature postnatal death in their transgenic mouse. This finding therefore suggests that the cause of immediate death in the Pu.1 mutant has multiple origins, albeit with breathing deficits very probably being the fatal contributor. We now address this specific aspect in a more detailed manner in a new paragraph in the Discussion (pages 13-14).

Thus, in summary, excellent paper but please add some more refs/discussion or data to underline that the deficits that you have shown are a major contributor to immediate postnatal death.ANY sign of Peripheral deficits; eg upper airway, heart, or lung anatomical /functional deficits?

This point is now addressed more fully in the Discussion, as stated above.